# Coarse-to-Fine Concept Bottleneck Models

**Konstantinos P. Panousis**[1,2,5*] **Dino Ienco**[1,2,3,4] **Diego Marcos**[1,2]

[1]Inria  [2]University of Montpellier  [3]Inrae  [4]UMR TETIS  [5]Department of Statistics, AUEB

panousis@aueb.gr
diego.marcos@inria.fr
dino.ienco@inrae.fr

## Abstract

Deep learning algorithms have recently gained significant attention due to their impressive performance. However, their high complexity and un-interpretable mode of operation hinders their confident deployment in real-world safety-critical tasks. This work targets *ante hoc* interpretability, and specifically Concept Bottleneck Models (CBMs). Our goal is to design a framework that admits a highly interpretable decision making process with respect to human understandable concepts, on *two levels of granularity*. To this end, we propose a novel two-level concept discovery formulation leveraging: (i) recent advances in vision-language models, and (ii) an innovative formulation for *coarse-to-fine concept selection* via data-driven and sparsity-inducing Bayesian arguments. Within this framework, concept information does not solely rely on the similarity between the *whole* image and general unstructured concepts; instead, we introduce the notion of *concept hierarchy* to uncover and exploit more granular concept information residing in patch-specific regions of the image scene. As we experimentally show, the proposed construction not only outperforms recent CBM approaches, but also yields a principled framework towards interpetability.

## 1 Introduction

The recent advent of foundation models has greatly popularized the deployment of deep learning approaches to a variety of tasks and applications. However, in most cases, deep architectures are treated in an alarming *black-box* manner: given an input, they produce a particular prediction, with their mode of operation and complexity preventing any potential investigation of their decision-making process. This property not only raises serious questions concerning their deployment in safety-critical applications, but at the same time, it could actively preclude their adoption in settings that could otherwise benefit societal advances, e.g., medical applications.

This conspicuous *shortcoming* of modern architectures has fortunately gained a lot of attention from the research community in recent years, expediting the design of novel frameworks towards deep learning interpretability. Within this frame of reference, there exist two core approaches: *ante* and *post* hoc. The latter aims to provide *explanations* to conventional pretrained models, e.g., Network Dissection [1], while the former aims to devise *inherently* interpretable models. In this context, Concept Bottleneck Models (CBMs) constitute one of the best-known approaches [9]; these comprise: (i) an intermediate Concept Bottleneck Layer (CBL), a layer whose neurons are tied to human understandable *concepts*, e.g., textual descriptions, followed by (ii) a linear decision layer. Thus, the final decision constitutes a linear combination of the CBL's concepts, leading to a more interpretable decision mechanism. However, typical CBM approaches are accompanied by four significant drawbacks: (i) they commonly require hand-annotated concepts, (ii) they usually exhibit

---

*Code available at: https://github.com/konpanousis/Coarse-To-Fine-CBMs

lower performance compared to their non-interpretable counterparts, (iii) their interpretability is substantially impaired due to the sheer amount of concepts that need to be analysed during inference, and (iv) they are not suited for tasks that require greater granularity.

The first drawback has been recently addressed by incorporating vision-language models in the CBM pipeline; instead of relying on a fixed concept set, images and text can be projected in the common embedding space and compared therein. Mechanisms to restore performance have also been proposed, e.g., residual fitting [25]. The remaining two limitations however, still pose a significant challenge.

Indeed, CBMs commonly rely on a large amount of concepts, usually proportional to the number of classes of the given task; with more complex datasets, thousands of concepts may be considered. Evidently, this renders the investigation of the decision making task an *arduous* and *unintuitive* process. In this context, some works aim to reduce the amount of used concepts by imposing sparsity constraints upon concept activation. Commonly, post-hoc class-wise sparsity methods are considered [22, 13]; however, these tend to restrict the number of concepts on a *per-class* basis, enforcing *ad hoc* application-specific sparsity/performance thresholds, greatly limiting the flexibility of concept activation for each example. Recently, a data-driven per-example discovery mechanism has been proposed in [17]; this leverages binary indicators that explicitly denote the relevance of each concept towards the downstream task on a per-example basis, thus allowing for greater flexibility.

Even though these approaches aim to address the problem of concept over-abundance, they do not consider ways to emphasize finer concept information that may be present in a given image; they still exclusively target similarity between concepts and the *whole image*. In this setting, localized, low-level concepts (e.g., shape or texture), are predicted from a representation of the whole image, potentially leading to the undesirable use of top-down relations. For instance, the model detects some high-level concept (e.g., elephant), resulting in associated lower-level concept activations (e.g., tusks, wrinkled skin) that may not even be actually be visible. This can further lead to significant concept omission, i.e., information potentially crucial for tasks that require greater granularity, e.g., fine-grained part discovery, or even cases where the input is susceptible to multiple interpretations.

Drawing inspiration from this inadequacy of CBM formulations, we introduce a novel coarse-to-fine paradigm that allows for discovering and capturing both *high* and *low* level concept information. We achieve this objective by devising an end-to-end trainable hierarchical construction; in this setting, we exploit both the whole image, as well as information residing in individual isolated regions of the image, i.e., specific patches, to achieve the downstream task. These levels are linked together by intuitive and principled arguments, allowing for information and context sharing between them, paving the way towards more interpretable models. We dub our approach *Coarse-to-Fine Concept Bottleneck Models* (CF-CBMs); in principle, our framework allows for arbitrarily deep hierarchies using different representations, e.g., super-pixels. Here, we focus on the two-level setting, as a proof of concept for the potency of the proposed framework. Our contributions can be summarized as:

- We introduce a novel interpretable hierarchical model that allows for coarse-to-fine concept discovery, exploiting finer details residing in patch-specific regions of an image.
- We propose a novel way of assessing the interpretation capacity of our model based on the Jaccard index between ground truth concepts and learned data-driven binary indicators.
- We experimentally show that CF-CBMs outperform other SOTA approaches classification-wise, while substantially improving interpretation capacity.

## 2  Related Work

**Concept Bottleneck Models.** Let us denote by $\mathcal{D} = \{\boldsymbol{X}_n, \hat{\boldsymbol{y}}_n\}_{n=1}^{N}$, a dataset comprising $N$ image/label pairs, where $\boldsymbol{X}_n \in \mathbb{R}^{I_H \times I_W \times c}$ and $\hat{\boldsymbol{y}}_n \in \{0, 1\}^C$. Within the context of CBMs, a *concept set* $A = \{a_1, \ldots, a_H\}$, comprising $H$ concepts, e.g., textual descriptions, is also considered; the main objective is to re-formulate the prediction process, constructing a *bottleneck* that relies upon the considered concepts, in an attempt to design inherently interpretable models. In this context, early works on concept-based models [11, 9], were severely limited by requiring an extensive hand-annotated dataset comprising all the used concepts. To enhance the reliability of predictions of diverse visual contexts, probabilistic approaches, such as ProbCBM [7], introduce the concept of *ambiguity*, allowing for capturing the uncertainty both in concept and class prediction. The appearance of vision-language models, chiefly CLIP [18], has mitigated the need for hand-annotated data, allowing to easily make use of thousands of concepts, followed by a linear operator on the concept similarity to solve

the downstream task [13, 24]. However, this generally means that all concepts may simultaneously contribute to a given prediction, rendering the analysis of concept contribution an arduous and counter-intuitive task, severely undermining the sought-after interpetability. This has led to methods that seek a sparse concept representation, either by design [12] or data-driven [17] perspectives.

**Concept-based Classification.** To discover the relations between images and attributes, vision-language models, e.g., CLIP [18], are typically considered. These comprise an image and a text encoder, denoted by $E_V(\cdot)$ and $E_T(\cdot)$ respectively, trained in a contrastive manner [20, 2] to learn a common embedding space. After training, we can then project any image and text in this common space and compute the similarity between their ($\ell_2$-normalized) embeddings. Assuming a concept set $A$, with $|A| = H$, the most commonly considered measure is the cosine similarity $\boldsymbol{S}$:

$$\boldsymbol{S} \propto E_V(\boldsymbol{X})E_T(A)^T \in \mathbb{R}^{N \times H} \tag{1}$$

This *similarity-based characterization* yields a unique representation for each image and has recently been exploited to design models with interpretable decision processes such as CBM-variants [25, 13] and Network Dissection approaches[14, 15]. Within this context, let us consider a $C$-class classification setting; by introducing a linear layer $\boldsymbol{W}_c \in \mathbb{R}^{C \times H}$, we can perform classification via the similarity representation $\boldsymbol{S}$. The output of such a network yields:

$$\boldsymbol{Y} = \boldsymbol{S}\boldsymbol{W}_c^T \in \mathbb{R}^{N \times C} \tag{2}$$

The image/text encoders are typically kept frozen; training only pertains to the weight matrix $\boldsymbol{W}_c$.

However, this formulation comes with a key deficit: it is by-design limited to the granularity of the concepts that it can potentially discover in any particular image. Indeed, VLMs are commonly trained to match concepts to the *whole image*; this can lead to a *loss of granularity*, that is, important details may be either omitted or considered irrelevant. Yet, in complex tasks such as fine-grained classification or in cases where the decision is ambiguous, this can potentially hinder both the downstream task, but also interpretability. In these settings, it is likely that any low-level information present is not exploited, hindering any potential low-level investigation on the network's process. Moreover, this approach considers the *entire concept set* to describe an input; this not only greatly limits the flexibility of the considered framework, but also renders the interpretation analyses questionable due to the sheer amount of concepts that need to be analysed during inference [19]. In this work, we take a step beyond the classical definition of CBMs and consider the setting of *coarse-to-fine* concept-based classification based on similarities between images, patches and concepts.

## 3 Coarse-to-fine CBM

To construct our proposed CF-CBM framework, we first introduce two distinct modeling *levels*: *High (H)* and *Low (L)*. The *High* level aims to model the whole image, while the *Low* level investigates and aggregates information stemming from localized regions. At the outset, we define these levels as separate modules that can be individually used towards a downstream task using some *independent* concepts sets $A_H$ and $A_L$; intuitively, the former should comprise descriptions that characterize the main scenes/objects in the considered dataset, e.g., an ImageNet class name, such as *Arctic Fox*, while the latter, descriptions that are inferrable from localized regions of the image, e.g., characteristics of parts of the animal or the background.

Then, we present the notion of *hierarchy* of concepts. Specifically, we introduce *interdependence* between the sets to capture information in a coarse-to-fine-grained manner; in this setting, the concepts $A_H$ encapsulate concepts that characterize each image, in turn determining the *allowed subset* of concepts from the low-level concept pool $A_L$. Essentially, the concepts in $A_H$ capture a holistic representation of the image, while the low-level, their sub-characteristics, delving deeper into patch-specific regions, aiming to uncover finer-grained information. Each level aims to achieve the given downstream task, while information sharing takes place between them as we describe next.

### 3.1 High Level Concept Discovery Module

For the formulation of the *High Level* view of our CF-CBM model, we consider: (i) the whole image, (ii) a set of $H$ concepts $A_H$, and exploit the definitions of concept-based classification, i.e., Eqs. (1)-

(2). To this end, we introduce a single linear layer with weights $\boldsymbol{W}_{Hc} \in \mathbb{R}^{C \times H}$, yielding:

$$\boldsymbol{S}_H \propto E_V(\boldsymbol{X}) E_T(A_H)^T \in \mathbb{R}^{N \times H}, \tag{3}$$

$$\boldsymbol{Y}_H = \boldsymbol{S}_H \boldsymbol{W}_{Hc}^T \in \mathbb{R}^{N \times C} \tag{4}$$

Evidently, this formulation, fails to take into account the relevance of each concept towards the downstream task or any information redundancy, since all the considered concepts are potentially used; this also limits its emerging interpretation capacity due to the large amount of concepts that need to be analysed during inference. To bypass this drawback, we consider a novel, data-driven mechanism for concept discovery based on auxiliary *binary* latent variables.

**Concept Discovery.** To discover the subset of high-level concepts present in each example, we introduce the auxiliary binary latent variables $\boldsymbol{Z}_H \in \{0, 1\}^{N \times H}$; these operate in an "on-off" fashion, indicating, for each example, if a given concept needs to be considered to achieve the downstream task, i.e., $[\boldsymbol{Z}_H]_{n,h} = 1$ if concept $h$ is *active* for example $n$, and 0 otherwise. The output of the network is now given by the inner product between the classification matrix $\boldsymbol{W}_{Hc}$ and the *discovered concepts* as dictated by the binary indicators $\boldsymbol{Z}_H$:

$$\boldsymbol{Y}_H = (\boldsymbol{Z}_H \cdot \boldsymbol{S}_H) \boldsymbol{W}_{Hc}^T \in \mathbb{R}^{N \times C} \tag{5}$$

A naive definition of these indicators would require computing and storing one indicator per example. To avoid the computational complexity and generalization limitations of such a formulation, we consider an *amortized* approach similar to [17]. To this end, we introduce a data-driven random sampling procedure for $\boldsymbol{Z}_H$, and postulate that the latent variables are drawn from appropriate Bernoulli distributions; specifically, their probabilities are proportional to a separate linear computation between the *embedding of the image* and an *auxiliary linear layer* with weights $\boldsymbol{W}_{Hs} \in \mathbb{R}^{H \times K}$, where $K$ is the dimensionality of the embedding, yielding:

$$q([\boldsymbol{Z}_H]_n) = \text{Bernoulli}\left([\boldsymbol{Z}_H]_n \Big| \text{sigmoid}\left(E_V(\boldsymbol{X}_n) \boldsymbol{W}_{Hs}^T\right)\right) \tag{6}$$

where $[\cdot]_n$ denotes the $n$-th row of the matrix, i.e., the indicators for the $n$-th image. This formulation exploits an *additional source of information* emerging solely from the image embedding; this allows for an *explicit* mechanism for inferring concept activation in the context of the considered task, instead of exclusively relying on the *implicit* VLM similarity. The described process is encapsulated in what we call a Concept Discovery Block (CBD); this is illustrated in Fig.1 (Left and Upper Right).

## 3.2 Low Level Concept Discovery Module

Here, we present a variant of the described architecture that aims to individually exploit finer information potentially present in the image. In this setting, re-using the whole image may hinder concept discovery since fine-grained details may be ignored; prominent objects may dominate the discovery task, especially in complex scenes, while omitting other significant attributes present in different regions of the image.

To facilitate the discovery of low-level information, avoiding conflicting information in the context of whole image, we split each image $n$ into a set of $P$ *non-overlapping* patches: $\boldsymbol{P}_n = \{\boldsymbol{P}_n^1, \dots, \boldsymbol{P}_n^P\}$, where $\boldsymbol{P}_n^p \in \mathbb{R}^{P_H \times P_W \times c}$ and $P_H, P_W$ denote the height and width of each patch respectively, and $c$ is the number of channels. In this context, each patch is now treated as a standalone image. To this end, we first compute their similarities with respect to a set of low-level concepts $A_L$. For each image $n$ split into $P$ patches, the patches-concepts similarity computation reads:

$$[\boldsymbol{S}_L]_n \propto E_V(\boldsymbol{P}_n) E_T(A_L)^T \in \mathbb{R}^{P \times L}, \quad \forall n \tag{7}$$

We define a single classification layer with weights $\boldsymbol{W}_{Lc} \in \mathbb{R}^{C \times L}$, while for obtaining a single representation vector for each image, we introduce an *aggregation* operation to combine the information from all the patches. This can be performed before or after the linear layer. Here, we consider the latter, using a maximum rationale. Thus, for each image $n$, the output $[\boldsymbol{Y}_L]_n \in \mathbb{R}^C$, reads:

$$[\boldsymbol{Y}_L]_n = \max_p \left[ [\boldsymbol{S}_L]_n \boldsymbol{W}_{Lc}^T \right]_p \in \mathbb{R}^C, \quad \forall n \tag{8}$$

where $[\cdot]_p$ denotes the $p$-th row of the matrix. Similar to the high-level, we define the corresponding concept discovery mechanism for the low level to address information redundancy; then, we introduce an information linkage between the different levels towards context sharing between them.

**Concept Discovery.** For each patch $p$ of image $n$, we consider latent variables $[\boldsymbol{Z}_L]_{n,p} \in \{0,1\}^L$, operating in an "on"-"off" fashion as before. Specifically, we introduce an amortization matrix $W_{Ls} \in \mathbb{R}^{L \times K}$, $K$ being the dimensionality of the embeddings. In this setting, $[\boldsymbol{Z}_L]_{n,p}$ are drawn from Bernoulli distributions driven from the patch embeddings, s.t.:

$$q([\boldsymbol{Z}_L]_{n,p}) = \text{Bernoulli}\left([\boldsymbol{Z}_L]_{n,p}\big|\text{sigmoid}\left(E_V([\boldsymbol{P}]_{n,p})W_{Ls}^T\right)\right) \tag{9}$$

The output is now given by the inner product between the *discovered low level concepts* as dictated by $\boldsymbol{Z}_L$ and the weight matrix $\boldsymbol{W}_{Lc}$, yielding:

$$[\boldsymbol{Y}_L]_n = \max_p \left[\left([\boldsymbol{Z}_L]_n \cdot [\boldsymbol{S}_L]_n\right)\boldsymbol{W}_{Lc}^T\right]_p \in \mathbb{R}^C, \; \forall n \tag{10}$$

The formulation of the low-level, patch-focused variant is now concluded. This module can be used as a standalone network to uncover information residing in patch-specific regions of an image and investigate the network's decision making process as shown in Fig.1 (Lower Right). However, by slightly altering its definition, we can straightforwardly introduce a linkage between the two described levels, allowing the flow of information between them.

### 3.3 Linking the two levels

For formulating a finer concept discovery mechanism, we introduce the notion of *concept hierarchy*. In this setting, we do not assume individual concepts sets, but instead posit that *each* high-level concept in $A_H$ is characterized by $L$ low-level attributes, such that $|A_L| = H \times L$. For tying the two different levels together, we augment the dimension of the low level module to account for this modification, and exploit the resulting latent variables $\boldsymbol{Z}_H$ and $\boldsymbol{Z}_L$ to devise a novel way of deciding which low-level attributes should be considered according to the active high level concepts dictated by $Z_H$; this allows for context exchange between the two levels and end-to-end training.

The underlying principle is that we can now *mask* the low-level concepts, i.e., *zero-out* the ones that are irrelevant, following a top-down rationale. During training, we learn which high-level concepts are active, and subsequently discover the essential low-level attributes, while the probabilistic nature of our construction allows for the consideration of different configurations of high and low level concepts. This leads to a rich information exchange between the levels of the network towards achieving the downstream task.

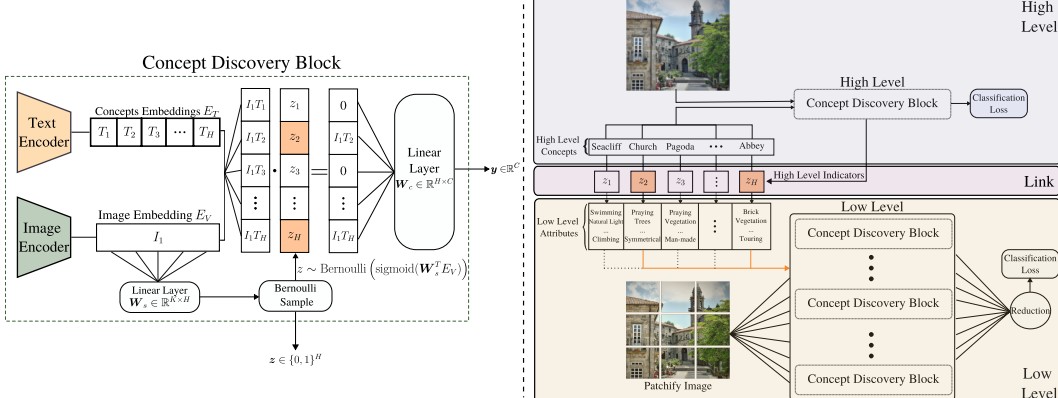

Figure 1: (Left) The Concept Discovery Block (CDB). Given a set of concepts and an image, we compute their similarity via a VLM; we consider a data-driven mechanism for concept discovery, sampling from an amortized Bernoulli posterior. (Right) A schematic of the envisioned CF-CBMs. We consider a set of high level concepts, each described by a number of attributes; this forms the *pool* of low-level concepts. Our objective is to discover concepts that describe the whole image, while exploiting information residing in, in this case $P = 9$, patch-specific regions by matching low-level concepts to each patch and aggregate the information to obtain a single representation. Each level comprises CDBs, while the levels are linked together via the binary indicators $\boldsymbol{Z}_H$ and $\boldsymbol{Z}_L$.

To formalize this linkage, we first consider which high-level concepts are active via $\boldsymbol{Z}_H$ to uncover which low-level attributes should be considered; then, we use the augmented indicators $\boldsymbol{Z}_L$ to further

mask the remaining low-level attributes according to the values therein. This yields:

$$[Z]_{n,p} \propto \sum_h [Z_H]_{n,h} \cdot [Z_L]_{n,p,h,:} \in \{0,1\}^L \tag{11}$$

Thus, by replacing the indicators $\boldsymbol{Z}_L$ in Eq. (10) with $\boldsymbol{Z}$, the two levels are linked together and can be trained in an end-to-end fashion. A graphical illustration of this linkage and the proposed Coarse-to-Fine CBM (CF-CBM) is depicted on Fig. 1 (Middle Right) and (Right) respectively. The introduced framework can easily accommodate more than two levels of hierarchy, while allowing for the usage of different input representations, e.g., super-pixels.

### 3.4 Training & Inference

**Training.** Considering a dataset $\mathcal{D} = \{(\boldsymbol{X}_n, \hat{\boldsymbol{y}}_n)\}_{n=1}^N$, we employ the standard cross-entropy loss, denoted by $\mathrm{CE}(\hat{\boldsymbol{y}}_n, f(\boldsymbol{X}_n, \boldsymbol{A}))$, where $f(\boldsymbol{X}_n, \boldsymbol{A}) = \mathrm{Softmax}([\boldsymbol{Y}]_n)$ are the class probabilities. For the simple concept-based model, i.e., without any discovery mechanism, the logits $[\boldsymbol{Y}]_n$ correspond to either $[\boldsymbol{Y}_H]_n$ (Eq.(4)), or $[\boldsymbol{Y}_L]_n$ (Eq.(8)), depending on the considered level. In this context, the only trainable parameters are the classification matrices for each level, i.e., $\boldsymbol{W}_{Hc}$ or $\boldsymbol{W}_{Lc}$.

For the full model, the presence of the indicator variables, i.e., $\boldsymbol{Z}_H$ and/or $\boldsymbol{Z}_L$, necessitates a different treatment of the objective. To this end, and in line with recent literature [16, 17], we turn to the Variational Bayesian (VB) framework, and specifically to Stochastic Gradient Variational Bayes (SGVB) [8]. We impose appropriate prior distributions on the latent indicators $\boldsymbol{Z}_H$ and $\boldsymbol{Z}_L$, s.t.:

$$\boldsymbol{Z}_H \sim \mathrm{Bernoulli}(\alpha_H), \qquad \boldsymbol{Z}_L \sim \mathrm{Bernoulli}(\alpha_L) \tag{12}$$

where $\alpha_H$ and $\alpha_L$ are non-negative constants. When the levels are linked together, the model comprises two outputs, and thus, the loss function consists of two distinct CE terms: (i) one for the high, and (ii) one for the low level. The objective function takes the form of an Evidence Lower Bound (ELBO)[5] provided in the Appendix. Obtaining the objective for a single level is trivial; one only needs to remove the other level's terms. For training, we turn to Monte Carlo (MC) sampling using a single reparameterized sample for each latent variable. Since, the Bernoulli is not amenable to the reparameterization trick [8], we turn to its continuous relaxation, i.e., the Gumbel-Softmax trick [10, 6]; we present the exact sampling procedure in the appendix.

**Inference.** After training, we can directly draw samples from the learned posteriors and perform inference. However, the stochastic nature of the indicators could potentially lead to multiple interpretations for the same input when drawing different samples. Commonly, there are two approaches to address this issue in the VB community: (i) draw multiple samples and average the results, or (ii) directly use the mean as an approximation to the aforementioned process; here, we opt for the latter. In our framework, the binary indicators are modeled via a Bernoulli distribution; thus, the mean corresponds to the probability of a concept being active for a particular example. Within this context, we can further introduce an interpretable threshold $\tau$; if the probability of a particular concept is greater than $\tau$, we consider it to be active, i.e., its indicator has a value of one, and zero otherwise. We use this formulation during inference, obtaining a single sparse interpretable representation for each image.

## 4 Experimental Evaluation

**Experimental Setup.** We consider three benchmark datasets for evaluating the proposed framework, namely, CUB[21], SUN[23], and ImageNet-1k[3]. These constitute highly diverse datasets varying in both number of examples and applicability: ImageNet is a 1000-class object recognition benchmark, SUN comprises 717 classes with a limited number of examples for each, while CUB is used for fine-grained bird species identification spanning 200 classes. For the VLM, we turn to CLIP [18] and select a common backbone, i.e., ViT-B/16. To avoid having to calculate the embeddings of both images/patches and text at each iteration, we pre-compute them with the chosen backbone. Then, during training, we load them and compute the necessary quantities. For the high level concepts, we consider the class names for each dataset. For the low-level concepts, for Imagenet, we randomly select 20 concepts for each class from the concept set described in [24] while for SUN and CUB, we exploit a per-class summary of the included attributes comprising 102 and 312 descriptions respectively. Since these are shared among classes and the number of active attributes differ for

Table 1: Classification Accuracy and Average Percentage of Activated Concepts (Sparsity). By **bold** blue/red, we denote the best-performing high/low level *sparsity*-inducing concept-based model.

| Architecture Type | Model | Concepts | Sparsity | Dataset (Accuracy (%) || Sparsity (%)) | | |
|---|---|---|---|---|---|---|
| | | | | CUB | SUN | ImageNet |
| Non-Interpretable | Baseline (Images) | ✗ | ✗ | 76.70 | 42.90 | 76.13 |
| | CLIP Embeddings[H] | ✗ | ✗ | 81.90 | 65.80 | 79.40 |
| | CLIP Embeddings[L] | ✗ | ✗ | 47.80 | 46.00 | 62.85 |
| Concept-Based Whole Image High Level | Label-Free CBMs | ✓ | ✓ | 74.59 | – | 71.98 |
| | CDM[H] | ✓ | ✗ | 80.30 | 66.25 | 75.22 |
| | CDM[H] | ✓ | ✓ | 78.90||19.00 | **64.55**||13.00 | 76.55||14.00 |
| | CF-CBM[H] (Ours) | ✓ | ✓ | **79.50**||50.00 | 64.00||47.58 | **77.40**||27.20 |
| Concept-Based Patches Low Level | CDM[L] | ✓ | ✗ | 39.05 | 37.00 | 49.20 |
| | CDM[L] | ✓ | ✓ | 59.62||58.00 | 42.30||67.00 | 58.20||25.60 |
| | CF-CBM[L] (Ours) | ✓ | ✓ | **73.20**||29.80 | **57.10**||28.33 | **78.45**||15.00 |

each class, we devise an efficient alternative linkage formulation to accommodate this setting; this is provided in the Appendix. These distinct sets enables us to assess the efficacy of the proposed framework in highly diverse configurations. We use $P = 16$ patches and set $\tau = 0.05$; this translates to a concept being active for the input only if it has probability of being active greater than 5%. We observed no significant variation when using smaller values. Larger values translate to fewer concepts being active, despite having significant activation probability before thresholding. We consider both classification accuracy, as well as the capacity of the proposed framework towards interpretability.

**Accuracy.** We begin our experimental analysis by assessing both the classification capacity of the proposed framework, but also its *concept sparsification* ability. To this end, we consider: (i) a baseline non-intepretable backbone, (ii) the recently proposed SOTA Label-Free CBMs [13], (iii) classification using only the clip embeddings either of the whole image (CLIP Embeddings[H]) or the image's patches (CLIP Embeddings[L]), (iv) classification based on the similarity between images and the *whole* concept set (CDM[H] ✗discovery), and (v) the approach of [17] that considers a data-driven concept discovery mechanism only on the whole image (CDM[H]✓discovery). We also consider the proposed patch-specific variant of CDMs defined in Sec. 3.2, denoted by CDM[L]. The baseline results and the Label-Free CBMs are taken directly from [13]. We denote our framework as CF-CBM.

In this setting, CLIP[H] and CDM[H] consider the concept set $A_H$, while patch-focused models, i.e., CLIP[L] and CDM[L], solely consider the low-level set $A_L$. Here, it is worth noting that the CDM[L] setting corresponds to a variant of the full CF-CBM model, where all the high level concepts are active; thus, all attributes are considered in the low-level with no masking involved. In this case, since the binary indicators $Z_H$ are not used, there is no information exchange taking place between the levels; this serves as an ablation setting of the impact of the information linkage. The obtained comparative results are depicted in Table 1. Therein, we observe that the proposed framework exhibits highly improved performance compared to Label-Free CBMs, while on par or even improved classification performance compared to the concept discovery-based CDMs on the high-level. On the low level, our approach improves performance up to $\approx 20\%$ compared to CDM[L].

At this point, it is important to highlight the effect of the hierarchical construction and the linkage of the levels to the overall behavior of the network. In all the considered settings, we observe: (i) a drastic improvement of the classification accuracy of the low-level module, and (ii) a significant change in the patterns of concept discovery on both levels. We posit that the information exchange that takes place between the levels, conveys a *context* of the relevant attributes that should be considered. This is reflected both to the capacity to improve the low-level classification rate compared to solely using the CLIP[L] or CDM[L], but also on the drastic change of the concept retention rate of the low level. At the same time, the patch-specific information discovered on the low-level alters the discovery patterns of the high-level, since potentially more concepts should be activated in order to successfully achieve the downstream task. This behavior is extremely highlighted in the ImageNet case: our approach not only exhibits significant gains compared to the alternative concept-based CDM on the high-level, but also the low-level accuracy of our approach *outperforms* it by a large margin. These first investigations hint at the capacity of the proposed framework to exploit patch-specific information for improving performance on the considered downstream task.

**Attribute Matching.** Even though classification performance constitutes an important indicator of the overall capacity of a given architecture, it is not an appropriate metric for quantifying its behavior within the context of interpretability. To this end, and contrary to recent approaches that

Table 2: Attribute matching accuracy. We compare our approach to the recent CDM model trained with the considered $A_L$ set. Then, we predict the matching between the inferred per-example concept indicators to: (i) class-wise and (ii) per-example ground truth attributes found in both SUN and CUB.

| Model | Attribute Set Train | Atrribute Set Eval | Dataset (Matching Accuracy (%)\|\| Jaccard Index (%)) | |
|---|---|---|---|---|
| | | | SUN | CUB |
| CDM[17] | whole set | class-wise | 51.43\|\|26.00 | 39.00\|\|17.20 |
| CDM$^L$ | whole set | class-wise | 30.95\|\|26.70 | 25.81\|\|19.60 |
| CF-CBM (Ours) | hierarchy | class-wise | **53.10\|\|28.20** | **79.85\|\|32.50** |
| CDM[17] | whole set | example-wise | 48.45\|\|15.70 | 36.15\|\|09.50 |
| CDM$^L$ | whole set | example-wise | 20.70\|\|15.00 | 17.65\|\|10.40 |
| CF-CBM (Ours) | hierarchy | example-wise | **49.92\|\|16.80** | **81.00\|\|17.60** |

solely rely on classification performance and qualitative analyses, we introduce a metric to measure the effectiveness of a concept-based approach. Thus, we turn to the *Jaccard Similarity* and compute the similarity between the binary indicators $z$ that denote the *discovered* concepts and the binary ground truth indicators that can be found in both CUB and SUN; we denote the latter as $z^{\text{gt}}$.

Let us denote by: (i) $M_{11}$ the number of entries equal to 1 in both binary vectors, (ii) $M_{0,1}$ the number of entries equal to 0 in $z$, but equal to 1 in $z^{\text{gt}}$, and (iii) $M_{1,0}$ the number of entries equal to 1 in $z$, but equal to 0 in $z^{\text{gt}}$; we consider the *asymmetric case*, focusing on the importance of correctly detecting the presence of a concept. Then, we can compute the Jaccard similarity as:

$$\text{Jaccard}(z, z^{\text{gt}}) = M_{1,1}/(M_{1,1} + M_{1,0} + M_{0,1}) \tag{13}$$

This metric can be exploited as an objective score for evaluating the quality of the obtained concept-based explanations across multiple frameworks, given that they consider the same concept set and ground truth indicators exist.

For a baseline comparison, we train a CDM with either: (i) the whole image (CDM) or (ii) the image patches (CDM$^L$), using the *whole set* of low-level attributes as the concept set for both SUN and CUB. We consider the same set for the low-level of CF-CBMs; due to its hierarchical nature however, CF-CBM exploits *concept hierarchy* as described in Sec.3.3 to narrow down the concepts considered on the low-level. For both SUN and CUB, we have ground truth attributes on a per-example basis (*example-wise*), but also the present attributes per class (*class-wise*). We assess the matching between these ground-truth indicators and the inferred indicators both in terms of binary accuracy, but also in terms of the considered Jaccard index.

In Table 2, the attribute matching results are depicted. Therein we observe, that our CF-CBMs outperform both CDM and CDM$^L$ in all the different configurations and in both the considered metrics with up to $15\%$ absolute improvement. These results suggest that by exploiting concept and representation hierarchy, we can uncover more relevant low-level information. However, it is also important to note how the binary accuracy metric can be quite misleading. Indeed, the ground truth indicators, particularly in CUB, are quite sparse; thus, if a model predicts that most concepts are not relevant, we yield very high binary accuracy. Fortunately though, the proposed metric can successfully address this false sense of confidence as a more appropriate measure for concept matching.

**Ablation Study.** One important aspect of the proposed CF-CBM framework is the impact of the considered number of patches and subsequently the number of Concept Discovery Blocks on the low level. To this end, we perform an ablation study on this parameter, using the CUB dataset. We consider five configurations: (i) $P = 4$, (ii) $P = 9$, (iii) $P = 16$, (iv) $P = 64$, and (v) $P = 256$. The obtained results are presented in Table 3. We observe that setting the number of patches to $16$ or $64$ yields a significant performance improvement in the attribute matching capability. In this context, it is very important to emphasize that despite the fact that classification wise, the performance is similar to other configurations, the capacity of the model towards interpretability is substantially different. Consequently, it is essential to have an objective measure of the interpretation capacity in the context of interpretability focused methods as the one proposed in this work.

**Qualitative Analysis.** For our qualitative analysis, we focus on the ImageNet-1k validation set; this decision was motivated by the fact that it is the only dataset where attribute matching could not be assessed due to the absence of ground-truth information. Thus, in Fig. 2, we selected a random class (*Sussex Spaniel*) and depict: (i) the 20 originally considered concepts and (ii) the results of the concept discovery. In this setting, we consider a concept to be relevant to the class if it is present in more than $40\%$ of the examples of the class; these concepts are obtained by averaging over the class

Table 3: An ablation study on the effect of the number of patches used on the low-level on the CUB dataset for both classification and interpretation capacity of the CF-CBM framework.

| $P$ | Acc. ‖ Spars. High (%) | Acc. ‖ Spars. Low (%) | Example-wise Jaccard (%) | Class-wise Jaccard (%) |
|---|---|---|---|---|
| 4 | 79.05 ‖ 55.00 | 73.70 ‖ 35.27 | 16.60 | 27.20 |
| 9 | 79.20 ‖ 47.80 | 72.00 ‖ 27.00 | 16.10 | 27.20 |
| 16 | 79.50 ‖ 50.00 | 73.20 ‖ 29.80 | 17.60 | **32.50** |
| 64 | 79.00 ‖ 47.60 | 73.40 ‖ 29.00 | **18.00** | 31.50 |
| 256 | 79.15 ‖ 47.60 | 73.40 ‖ 25.00 | 16.70 | 27.40 |

examples' indicators. We observe that CF-CBM is able to retain highly relevant concepts from the original set, while discovering equally relevant concepts from other classes such as *australian terrier, soft-coated wheaten terrier* and *collie*.

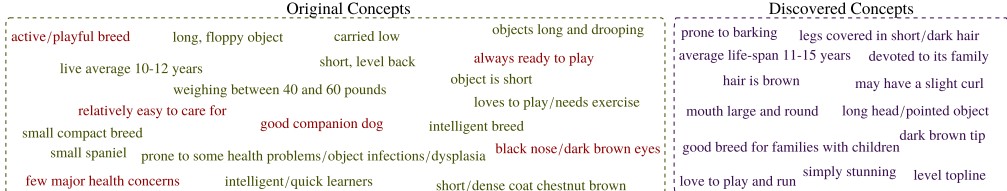

Figure 2: Original and additional discovered concepts for the *Sussex Spaniel* ImageNet class. By green, we denote the concepts retained from the original low-level set pertaining to the class, by maroon, concepts removed via the binary indicators $Z$, and by purple, the newly discovered concepts.

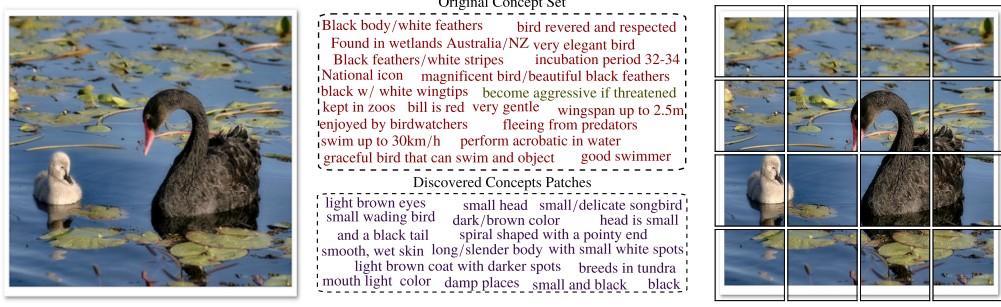

Figure 3: A random example from the *Black Swan* class of ImageNet-1k validation set. On the upper part, the original concept set corresponding to the class is depicted; on the lower, some of the concepts discovered via our novel CF-CBM.

In Fig.3, we focus on the example-wise behavior of the proposed framework. To this end, and for a random image from the ImageNet-1k validation set, we illustrate: (i) the original low-level set of attributes describing its class (*Black Swan*), and (ii) some of the low-level attributes discovered by our CF-CBM. We observe that the original concept set pertaining to the class cannot adequately represent the considered example. Indeed, most concepts therein would make the interpretation task difficult even for a human annotator. In stark contrast, the proposed framework allows for a more interpretable set of concepts, capturing finer information residing in the patches; this can in turn facilitate a more thorough examination of the network's decision making process.

In this context, and since we have access to all the concepts/attributes discovered on the image and patch level, we can examine and visualize the discovered concepts for each patch and assess their validity. We provide such a visualization in Fig. 4, where for the *Black Swan* image of Fig.3, we present five of the most contributing discovered attributes on the patch-wise level, i.e., attributes that are inferred to be active for this example, encoded in the binary masks $Z$, and that have a large contribution to the classification decision. Therein, we observe that our approach is able to exploit the information residing in localized regions of the image, granting significant insights towards the decision-making process of the proposed framework.

Additional qualitative investigations with respect to the inferred concept patterns are provided in the Appendix.

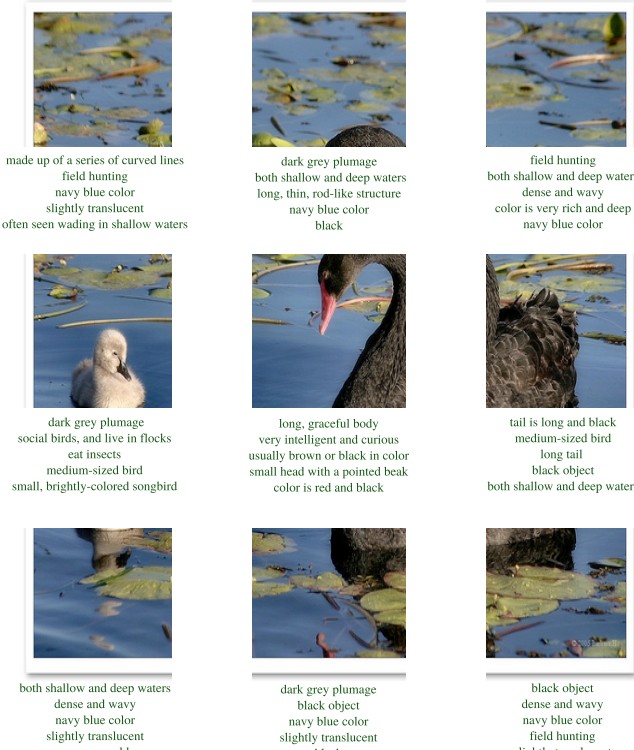

Figure 4: Five of the most activated concepts per patch for a Black Swan image from the ImageNet validation set. After training, we have access to the full set of activated concepts for each patch allowing the examination of the inferred active attributes. This facilitates the examination of their validity, while providing insights for the decision making process of the proposed CBM-based framework.

## 5 Limitations & Conclusions

A potential limitation of the CF-CBM framework is the dependence on the vision-language backbone. The final performance and interpretation capacity is tied to its suitability with respect to the task at hand. If the embeddings cannot adequately capture the relation (in terms of similarity) between images/patches-concepts, there is currently no mechanism to mitigate this issue. However, our construction easily accommodates adapting the backbone. Concerning the complexity of the proposed CF-CBM framework, by precomputing all the required embeddings, the resulting complexity is orders of magnitude lower than training a conventional backbone. A more thorough discussion on the limitations and complexity of our approach is presented in the Appendix.

In this work, we proposed an innovative framework in the context of ante-hoc interpretability based on a novel hierarchical construction. We introduced the notion of *concept hierarchy*, in which, high-level concepts are characterized by a number of lower-level attributes. In this context, we leveraged recent advances in CBMs and Bayesian arguments to construct an end-to-end coarse-to-fine network that can exploit these distinct concept representations, by considering both the whole image, as well as its individual patches; this facilitated the discovery and exploitation of finer information residing in patch-specific regions of the image. We validated our paradigm both in terms of classification performance, while considering a new metric for evaluating the network's capacity towards interpretability. As we experimentally showed, we yielded networks that retain or even improve classification accuracy, while allowing for a more fine-grained investigation of their decision process.

## Acknowledgements

This work was supported by ANR project OBTEA ANR-22-CPJ1-0054-01.

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

# Appendix

## A Training and Experimental Details

Considering a dataset $\mathcal{D} = \{(\boldsymbol{X}_n, \hat{\boldsymbol{y}}_n)\}_{n=1}^N$, we employ the standard cross-entropy loss, denoted by $\text{CE}(\hat{\boldsymbol{y}}_n, f(\boldsymbol{X}_n, \boldsymbol{A}))$, where $f(\boldsymbol{X}_n, \boldsymbol{A}) = \text{Softmax}([\boldsymbol{Y}]_n)$ are the class probabilities. For the simple concept-based model, i.e., without any discovery mechanism, the logits $[\boldsymbol{Y}]_n$ correspond to either $[\boldsymbol{Y}_H]_n$ (Eq.(4)), or $[\boldsymbol{Y}_L]_n$ (Eq.(8)), depending on the considered level. In this context, the only trainable parameters are the classification matrices for each level, i.e., $\boldsymbol{W}_{Hc}$ or $\boldsymbol{W}_{Lc}$.

For the full model, the presence of the indicator variables, i.e., $\boldsymbol{Z}_H$ and/or $\boldsymbol{Z}_L$, necessitates a different treatment of the objective. To this end, we turn to the Variational Bayesian (VB) framework, and specifically to Stochastic Gradient Variational Bayes (SGVB) [8].

In the following, we consider the case where the levels are linked together. Obtaining the objective for a single level is trivial; one only needs to remove the other level's terms. Since the network comprises two outputs, and the loss function consists of two distinct CE terms: (i) one for the high-level, and (ii) one for the low-level. The final objective function takes the form of an Evidence Lower Bound (ELBO) [5]:

$$\mathcal{L}_{\text{ELBO}} = \sum_{i=1}^N \text{CE}(\hat{\boldsymbol{y}}_n, f(\boldsymbol{X}_n, \boldsymbol{A}_H, [\boldsymbol{Z}_H]_n)) + \text{CE}(\hat{\boldsymbol{y}}_n, f(\boldsymbol{X}_n, \boldsymbol{A}_L, [\boldsymbol{Z}]_n))$$
$$- \beta(\text{D}_{KL}(q([\boldsymbol{Z}_H]_n)||p([\boldsymbol{Z}_H]_n)) + \sum_p \text{D}_{KL}(q([\boldsymbol{Z}_L]_{n,p})||p([\boldsymbol{Z}_L]_{n,p}))), \tag{14}$$

where we augmented the CE notation to reflect the dependence on the binary indicators. $\beta$ is a scaling factor [4] to avert the KL term from dominating the downstream task. The KL term encourages the posterior to be close to the prior; setting $\alpha_H, \alpha_L$ to a very small value "pushes" the posterior to sparser solutions. Through training, we aim to learn which of these components effectively contribute to the downstream task.

For computing Eq. (14), we turn to Monte Carlo (MC) sampling using a single reparameterized sample for each latent variable. Since, the Bernoulli is not amenable to the reparameterization trick [8], we turn to its continuous relaxation using the Gumbel-Softmax trick [10, 6].

### A.1 Bernoulli Relaxation

Let us denote by $\tilde{\boldsymbol{z}}_i$, the probabilities of $q(\boldsymbol{z}_i)$, $i = 1, \dots N$. We can directly draw reparameterized samples $\hat{\boldsymbol{z}}_i \in (0, 1)^M$ from the continuous relaxation as:

$$\hat{\boldsymbol{z}}_i = \frac{1}{1 + \exp\left(-(\log \tilde{\boldsymbol{z}}_i + L)/\tau\right)} \tag{15}$$

where $L \in \mathbb{R}$ denotes samples from the Logistic function, such that:

$$L = \log U - \log(1 - U), \quad U \sim \text{Uniform}(0, 1) \tag{16}$$

where $\tau$ is called the *temperature* parameter; this controls the degree of the approximation: the higher the value the more uniform the produced samples and vice versa. We set $\tau$ to 0.1 in all the experimental evaluations. During inference, we can use the Bernoulli distribution to draw samples and directly compute the binary indicators.

### A.2 Experimental Details

For our experiments, we set $\alpha_H = \alpha_L = \beta = 10^{-4}$; we select the best performing learning rate among $\{10^{-4}, 10^{-3}, 5 \cdot 10^{-3}, 10^{-2}\}$ for the linear classification layer. We set a higher learning rate for $\boldsymbol{W}_{Hs}$ and $\boldsymbol{W}_{Ls}$ ($10\times$) to facilitate learning of the discovery mechanism.

For all our experiments, we use the Adam optimizer without any complicated learning rate annealing schemes. We trained our models using a single NVIDIA A5000 GPU with no data parallelization. For all our experiments, we split the images into $P = 16$ patches. The patch level CLIP embeddings are extracted by resizing first and passing through CLIP to match the standard CLIP input size. To

this end, we first resize the patch to 224. For SUN and CUB, we train the model for a maximum of 1000 epochs, while for ImageNet, we only train for 100 epochs.

For the baseline non-interpretable backbone, we follow the setup of LF-CBM[13]. Thus, for CUB and SUN we consider a RN18 model and for ImageNet, a RN50.

**Complexity.** For SUN and CUB, training each configuration for 1000 epochs, takes approximately 10 minutes (wall time measurement), while for ImageNet, 100 epochs require approximately 4 hours.

**Variability Investigation.** Given the probabilistic nature of our proposed framework, we investigate the factors of variability in the experimental results. In the context of CF-CBMs, these factors pertain to both the initialization of the parameters of the network and the random drawing of samples for the concept presence indicators throughout training. Thus, by performing multiple runs under given experimental conditions, while utilizing different seeds, we can confidently evaluate the variability. We consider the CUB dataset with $P = 16$ patches and perform ten different runs, each with a different random seed. In Table 4, the obtained results are depicted; therein, the mean classification accuracy and mean sparsity rate across the ten different runs are reported, along with the computed standard deviation among them. We report both the high and low level values. As we observe, the results are highly consistent, resulting in a negligible standard deviation for the classification rates and relatively small sparsity standard deviations $1.00$ and $1.40$ for the high and low level respectively.

Table 4: Mean accuracy, Mean Sparsity and their standard deviations for both the high and the low level of our CF-CBM framework. We performed ten different runs under different seeds to assess the effect of initialization and random sampling.

| Level | Mean Accuracy (%) | Standard Deviation | Mean Sparsity (%) | Standard Deviation |
|-------|-------------------|--------------------|-------------------|--------------------|
| High  | 79.50             | 0.15               | 52.00             | 1.00               |
| Low   | 73.60             | 0.22               | 31.60             | 1.40               |

**Sparsity Effect Ablation.** To evaluate the effect of the sparsity inducing mechanism on the obtained classification and interpretability results, we consider an ablation study, where we vary the Bernoulli prior probabilities $\alpha_H, \alpha_L$ that directly affect the obtained sparsity. The higher the value of each respective $\alpha$, the less sparse are the obtained results; this allows for assessing the impact of the sparsity inducing behavior of the proposed model. The results are depicted in Table 5; in the performed study, we consider the same value for $\alpha_H$ and $\alpha_L$, denoted as $\alpha$. Therein, we observe that the sparser the representation, the better the attribute matching capabilities of the model, while at the same time the classification performance is retained or even improves compared to a less sparse setting. These results also highlight the necessity of another metric apart from the classification accuracy to assess the interpretation capabilities of the resulting models, since all settings exhibit similar performance.

Table 5: An ablation study on the impact of the sparsity level in the resulting accuracy and Jaccard Similarity. We vary the values $\alpha_H, \alpha_L$ of the Bernoulli priors (using the same value denoted as $\alpha$); lower values lead to sparser representations, while values close to one lead to denser discovered concept sets.

| $\alpha$ | Accuracy (%) \|\| Sparsity (%) | | Jaccard Similarity | |
|----------|------------|-----------|------------|--------------|
|          | High Level | Low Level | Class Wise | Example Wise |
| 0.0001   | **79.50\|\|50.00** | **73.20\|\|29.80** | **32.50** | **17.60** |
| 0.001    | 79.00\|\|55.50 | 72.70\|\|31.10 | 29.20 | 16.60 |
| 0.01     | 78.75\|\|58.20 | 73.00\|\|32.20 | 26.70 | 16.60 |
| 0.1      | 79.00\|\|62.00 | 73.00\|\|37.00 | 26.10 | 16.50 |
| 1        | 77.60\|\|83.70 | 69.15\|\|43.70 | 25.00 | 15.60 |

**Alternative linkage formulation.** In the setting where high level concepts share some sub-characteristics, we can devise a more efficient way to mask the low-level concepts and reduce the computational complexity of the approach. Specifically, we begin by considering $H$ high level concepts in $A_H$ and a total of $L^{\text{all}}$ low level concepts in the $A_L$ concept set, which now comprises a concatenation of the sub-characteristics of all high level concepts. Evidently, if no sub-characteristics are shared by the high level concepts, we get $L^{\text{all}} = H \times L$ as in the general case presented in the

main text, otherwise, $L^{\text{all}} < H \times L$. This formulation can also accomodate concepts sets where all concepts are shared and each class has a different number of potential sub-characteristics, as is the case for CUB and SUN, where $L^{\text{all}}$ is 312 and 102 respectively.

In both cases though, since we know which sub-characteristics characterize each high-level concept $h$, we can consider a *fixed* $L$-sized binary vector $\boldsymbol{b}_h \in \{0,1\}^L$ that encodes this relationship; these are concatenated to form the matrix $\boldsymbol{B} \in \{0,1\}^{L \times H}$. Each entry $l, h$ therein, denotes if the low-level attribute $l$ characterizes the high-level concept $h$; if so, $[\boldsymbol{B}]_{l,h} = 1$, otherwise $[\boldsymbol{B}]_{l,h} = 0$.

To exploit this representation to formalize the linkage between the High and the Low level, we first consider which high-level concepts are active via $\boldsymbol{Z}_H$, and use $\boldsymbol{B}$ to uncover which low-level attributes should be considered in the final decision; this is computed via a mean operation, averaging over the high-level dimension $H$. Then, we use the indicators $\boldsymbol{Z}_L$ to further mask the remaining low-level attributes. This yields:

$$\boldsymbol{Z} \propto \left( \boldsymbol{Z}_H \boldsymbol{B}^T \right) \cdot \boldsymbol{Z}_L \tag{17}$$

Thus, by replacing the indicators $\boldsymbol{Z}_L$ in Eq.10 with $\boldsymbol{Z}$, the two levels are linked together and can be trained on an end-to-end fashion as before.

# B Discussion: Limitations

**Generalizability**: The focus on VLMs might limit the generalizability to other domains and data.

To the best of our knowledge, most (if not all) CBM models deal with the same type of data, and we could say that they share similar shortfalls. Evidently, our approach is centered around vision and language models. However, we posit that the consideration of two (or more) different kind of representations/views through our construction, instead of limiting the generalizability of the approach, it improves it since we can now exploit this structure to address cases where CBMs were lacking. This includes for example cases of Remote Sensing data, where one could have both satellite and ground level images, along with textual descriptions for each. This could potentially be generalized to all kinds of data, considering that multimodal models for multiple modalities are available. As long as we can project the data in a common embedding space and compute the similarities between modalities, we posit that our CF-CBM method is more expressive than other conventional CBM methods due to their structured construction. Nevertheless, assessing the applicability of our approach to other domains is indeed an important part of our future work.

**Complexity**: How the computationally heavy is the proposed framework?

Compared to other approaches that train or fine-tune conventional backbones or resort to complicated training schemes, our approach is relatively lightweight. Specifically, a large amount of the overhead relates to the computation of the CLIP embeddings, a process that takes a couple of minutes for each dataset. These are calculated once and from then on, we can use these embeddings as is; at the same time, training comprises learning just the linear layers present in the construction. The computational complexity in this setting is limited, considering we can run 1000 epochs for CUB/SUN in approximately 7 minutes on a single GPU for $P = 4$ and $P = 9$, 20 minutes for $P = 64$, and 30 minutes for $P = 256$ patches. We did not use any parallelization for our computations; this could help speeding up the process in case of even larger grids. Inference time is negligible, considering that our approach essentially comprises just inner products in a small number of layers.

During inference and for real-time projection into the common embedding space, there is substantial work that is taking place with respect to reducing the overhead in Multimodal and Large Language Models via quantization and other techniques. This would easily allow for on-line projection and estimation of the corresponding values even on low-power commodity devices such as smartphones.

**Dependence on the Pretrained Models**: The issue of the dependence to the pretrained models is indeed an important aspect of the approach.

As we briefly touch upon in the limitations section, the dependence on the considered vision-language backbone is very important for the final performance of the model. Mitigating this issue is a very challenging task; we need to somehow be able to compute the relation between the considered modalities in terms of either similarity or some other metric. Thus, we are tied to the usage of a backbone that projects the data in a common embedding space or where the representations are

aligned. CLIP and other contrastively-trained multimodal models at this point constitute the most easy to apply and intuitive frameworks. To enhance performance in a case where the backbone is not suitable for the considered task, one potential solution is to finetune the pretrained model on the task and the other is to train such a cross-modal model from scratch either deterministically or probabilistically (that could potentially allow for some uncertainty analysis and insights into the predictions). However, both solutions require ground truth data that may be difficult to obtain depending on the considered task. We aim to explore these possibilities in the recent future.

## C    The Top-down and Bottom-up view of concept design

As noted in the main text, in this work, we consider both high and low level concepts. Within this frame of reference, one design choice that needed to be addressed is the interpretation of the connection between these two sets. In our view, there are two ways to approach this setting: (i) top-down (which we consider), and (ii) bottom-up. We posit that a combined bottom-up-then-top-down approach is what would most closely follow a human-like behavior when analysing an object. However, it is the second step that is more of a conscious process: we first become aware of the whole object, e.g., a bird or a dog, even if we have subconsciously perceived a lot of low-level cues to reach that conclusion, and then, based on this high-level knowledge, we can draw further conclusions about the nature of the lower-level image characteristics, e.g. interpreting a furry texture as either feathers or fur.

In a purely bottom-up approach, we would first analyse the low-level characteristics, such as shapes and textures, and we would then try to reason about the whole context in order to assign them semantics, e.g. ears, tail, fur. In our opinion, there isn't a single right approach for solving this problem in the context of interpretability. We posit however, that the information exchange that takes places between the high and the low levels via the learning process of the binary indicators does indeed allows for context information sharing between both levels (in the forward pass only from high to low, but also the inverse during training).

One of the motivations of this work was to be able to examine not only the high level concepts but mainly the low level ones. This could potentially allow for drawing conclusions about the high level concept in terms of the uncovered low level attributes. In this context, we can focus on the discovered low-level attributes themselves and reason on the high-level concepts. In our opinion, this is somewhat captured in the proposed framework. Indeed, in the qualitative analyses, we observed that, many times, the discovered low level concepts revealed attributes that are semantically connected to various high level concepts.

**Future work.**    In our setting, we assume that there exists a known hierarchy/relationship between the concepts. However, it very well be the case that there exists some hidden/latent hierarchy in the ground truth attributes that is not explicitly captured via the construction of the concepts sets. In this case, an interesting extension to our proposed framework would be a compositional bottom-up approach with no a priori known hierarchies. Within this context, we could potentially devise a method that explicitly integrates the aforementioned bottom-up view, aiming to uncover the hidden hierarchies.

Finally, in this work, we opted for the simplest split of the image into non-overlapping regions, namely non-overlapping fixed-size patches. This modeling decision allowed for demonstrating the impact of the hierarchical construction towards finer concept discovery, while at the same time facilitated the investigation of the contribution of each individual level when considering a single general purpose vision language model. To augment the modeling capacity of the proposed framework, more involved splitting procedures can be considered for the low level, such as superpixels or other specific vision-language backbones such as Region-CLIP [26]. We aim to explore these avenues in the near future.

# D  Further Investigations and Qualitative Analyses

## D.1  Alignment between CLIP similarities and Concept Presence Indicators

To further investigate the behavior of the proposed framework, we examine the alignment between CLIP similarities and the inferred concept presence indicators; this allows for assessing if the inferred indicators only activate the most similar (in the CLIP sense) concepts.

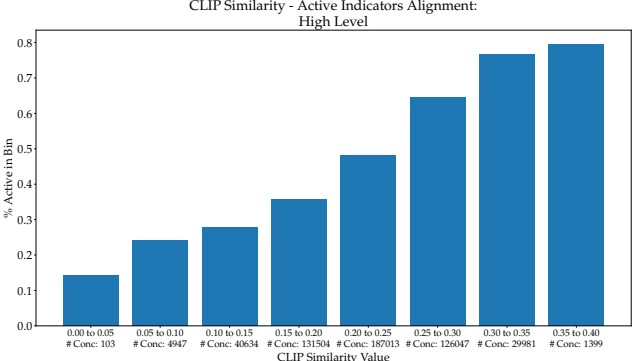

Figure 5: Alignment between the inferred concept presence indicators and CLIP similarities on the High Level of the CF-CBM framework. We split the CLIP similarities into bins of size $0.05$; in each bin we count the number of concepts assigned therein (according to their CLIP similarity and denoted by #Conc) and we compute the fraction of inferred active concepts to said number. We observe that in this case, the higher the similarity, concepts with high similarity value exhibit a largest percentage of activation.

For this exploration, we focus on the CUB dataset, split the CLIP similarities into bins of size $0.05$ and compute the ratio of activated concepts to the total number of concepts across all the examples of the validation set in each bin. The obtained statistics for the High Level are depicted in Fig. 5. We observe that on average, concepts with high CLIP similarity exhibit a larger percentage of activation in this level. Investigating this behavior on individual examples, e.g., Fig.6, reveals the same trend; however, even though we overall observed the same trend in multiple examples, at the same time, we can also observe the flexibility of the per-instance selection, where in several examples, concepts that have lower CLIP similarity values can also be active, even having examples with low CLIP similarity values more active than high CLIP similarity concepts as in Fig. 7.

The corresponding results for the Low level are depicted in Table 8. In contrast to the high level, we observe that on average, the concept activations with intermediate CLIP similarity values are more active. We posit that the high level can potentially exploit the information provided via the high

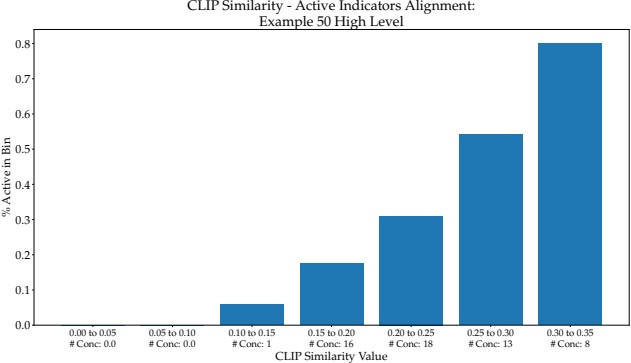

Figure 6: Alignment between the inferred concept presence indicators and CLIP similarities on the High Level of the CF-CBM framework for Example 50 in the CUB validation set. In this instance, we observe the same pattern as in Fig. 5.

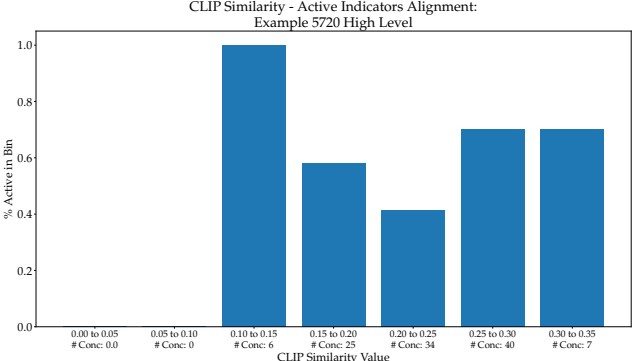

Figure 7: Alignment between the inferred concept presence indicators and CLIP similarities on the High Level of the CF-CBM framework for Example 5720 in the CUB validation set. In this instance, and contrary to the previous illustrations, we observe a different trend, where concepts with low-similarity value exhibit largest percentage of activation compared to concepts with high similarity.

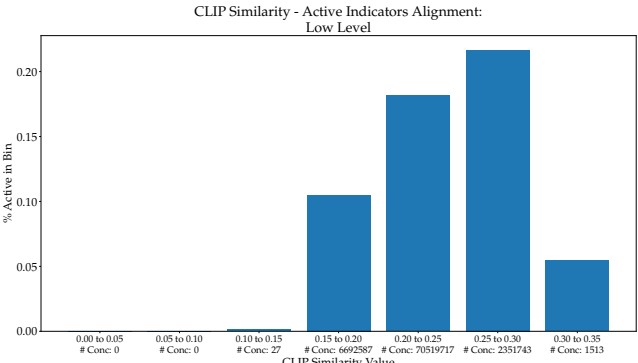

Figure 8: Alignment between the inferred concept presence indicators and CLIP similarities on the Low Level of the CF-CBM framework. We split the CLIP similarities into bins of size $0.05$; in each bin we count the number of concepts assigned therein (according to their CLIP similarity and denoted by #Conc) and we compute the fraction of inferred active concepts to said number. We observe that in this case, concepts with average similarity values have a higher percentage of activation compared to the concepts with the highest similarity.

level concepts, using the most similar ones, i.e., the ones that have high similarity with each instance, towards the classification task. On the contrary, on the low-level, the given concepts may very well describe several different patches; thus, the discovery mechanism aims to compensate the lack of meaningful classification signal through the diversification of the used concepts; this results in the utilization of concepts that have a lower -CLIP based- similarity in order to achieve classification. Nevertheless, the flexibility of the proposed framework allows specific examples to deviate from this trend, and individually exhibit distinct concept activation patterns to achieve the downstream task, e.g., Fig. 9.

### D.2 Concept Activation Patterns

As already discussed, one important aspect of the proposed CF-CBM framework concerns the flexibility of the per-example presence indication mechanism. In stark contrast to other commonly used sparsity-inducing approaches, the considered concept discovery mechanism allows for inferring the essential number and combination of active concepts towards achieving the downstream task. In turn, this leads to unique patterns of activations on both the individual examples but consequently on a class-wise level. To explore the emerging patterns, we plot the sum of active indicators of all examples in a class with respect to the considered concepts. The results for the High Level for

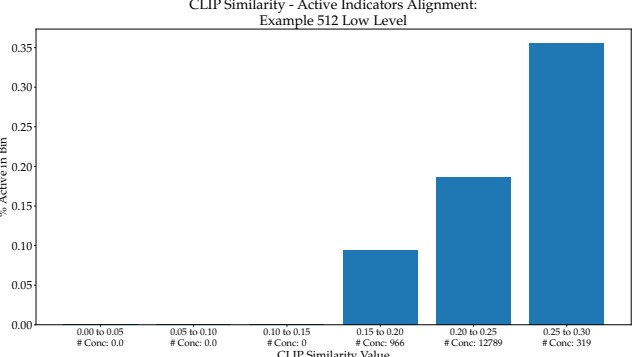

Figure 9: Alignment between the inferred concept presence indicators and CLIP similarities on the Lowe Level of the CF-CBM framework for Example 512 in the CUB validation set. In this instance, and contrary to the average pattern in the low level, we observe that the concepts with the highest similarity value exhibit the largest percentage of activation.

four random classes for the CUB dataset are illustrated in Fig. 10. We readily observe that each class yields a distinct pattern of concept activation. Within each class, there are several concepts are shared by a number of examples as one would expect; at the same time however, we observe how the data-driven concept discovery mechanism allows individuals examples to yield different activation patterns. This leads to concepts that are only active in one or two examples of the class, which were inferred nevertheless essential for their representation. We observe the same behavior in the low-level of the CF-CBM framework, depicted in Fig.11. Despite yielding a highly sparse representation, we observe distinct patterns of concept activations per class, retaining nevertheless the flexibility of modeling individual examples.

## D.3 Discussion: Differences with $\ell_1$ sparsity.

One of the most commonly used sparsity inducing methods is $\ell_1$. However, in our view, it is very restrictive in the context of interpretability. This regularization is typically imposed on the weights of the linear layer of a concept-based model; this would result into the -unwanted- effect of turning off concepts completely for all images or on a per class basis using more complicated solvers. At the same time, it typically requires some kind of ad-hoc and unintuitive sparsity-accuracy trade-off, while the solvers are typically applied in a post-hoc manner, e.g. [22, 13]. This greatly limits the flexibility of the approach, since either all images in general or all images in a particular class must follow the "global" or "class" pattern found by the solver, potentially omitting important information present in each individual image. On the other hand, we consider a per-instance concept discovery mechanism that is based on a simple data-driven Bernoulli distribution, which can explicitly denote concept relevance without confining the results in class or global representation, while training can be performed end-to-end using Stochastic Gradient Variational Bayes.

In this context, and as discussed in Section D.2, there are many cases where concepts do not contribute to a specific class at all, i.e., no instance of the class activated said concepts, potentially yielding similar conclusions to $\ell_1$-based methods. However, we also observed cases where a single instance in the class was using a particular concept. The same was also true for concepts active for only two or three instances. Within this context, we expect that the use of the conventional $\ell_1$-sparsity loss in a post hoc manner, along with the requirement for ad-hoc sparsity/performance thresholds would eliminate this within-class information; at the same time, applying such an $\ell_1$ based approach to a framework comprising two different connected levels and subsequently two distinct classification losses, at least in a principled way, is at this point, not clear.

## D.4 Further Qualitative Analysis

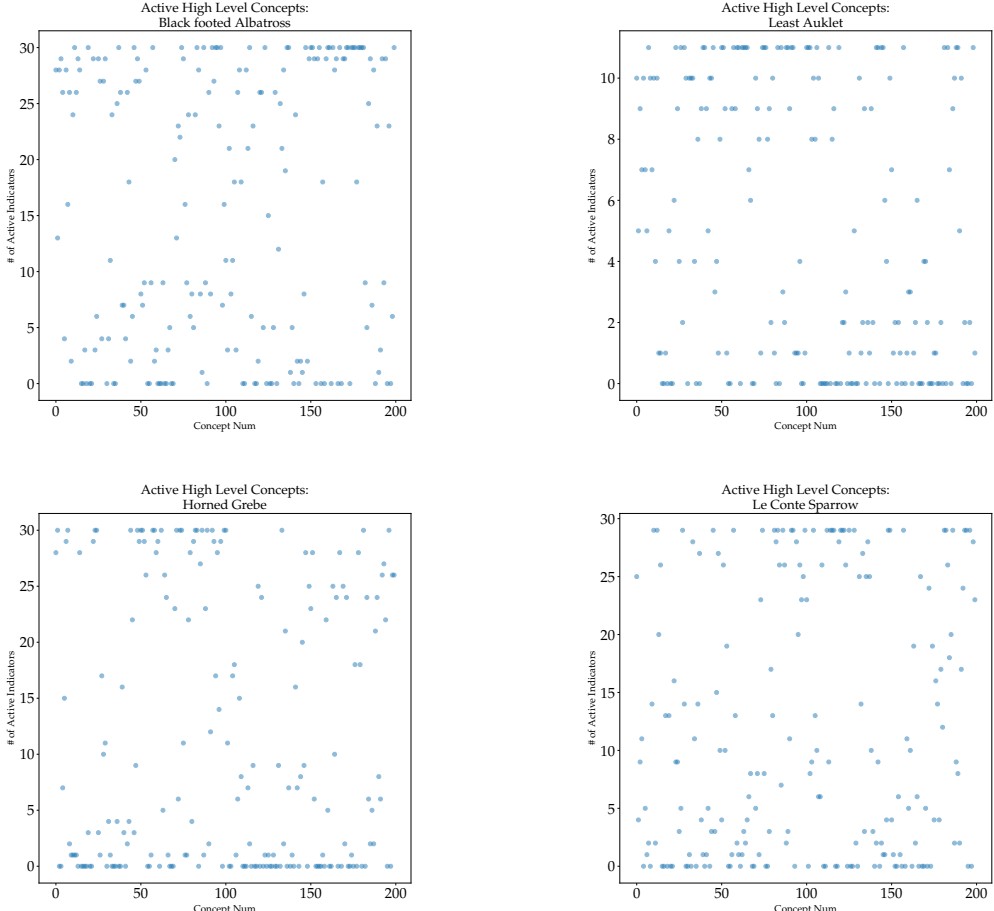

Figure 10: Summary of Activation of High Level concepts for Classes: *Black Footed Albatross, Least Auklet, Horned Grebe* and *Le Conte Sparrow*. We readily observe that each class yields a different pattern of concept activation. Within each class, there are several concepts are shared by a number of examples as one would expect; at the same time however, we observe how the data-driven concept discovery mechanism allows individuals examples to yield different activation patterns. This leads to concepts that are only active in one or two examples of the class, which were inferred nevertheless essential for their representation.

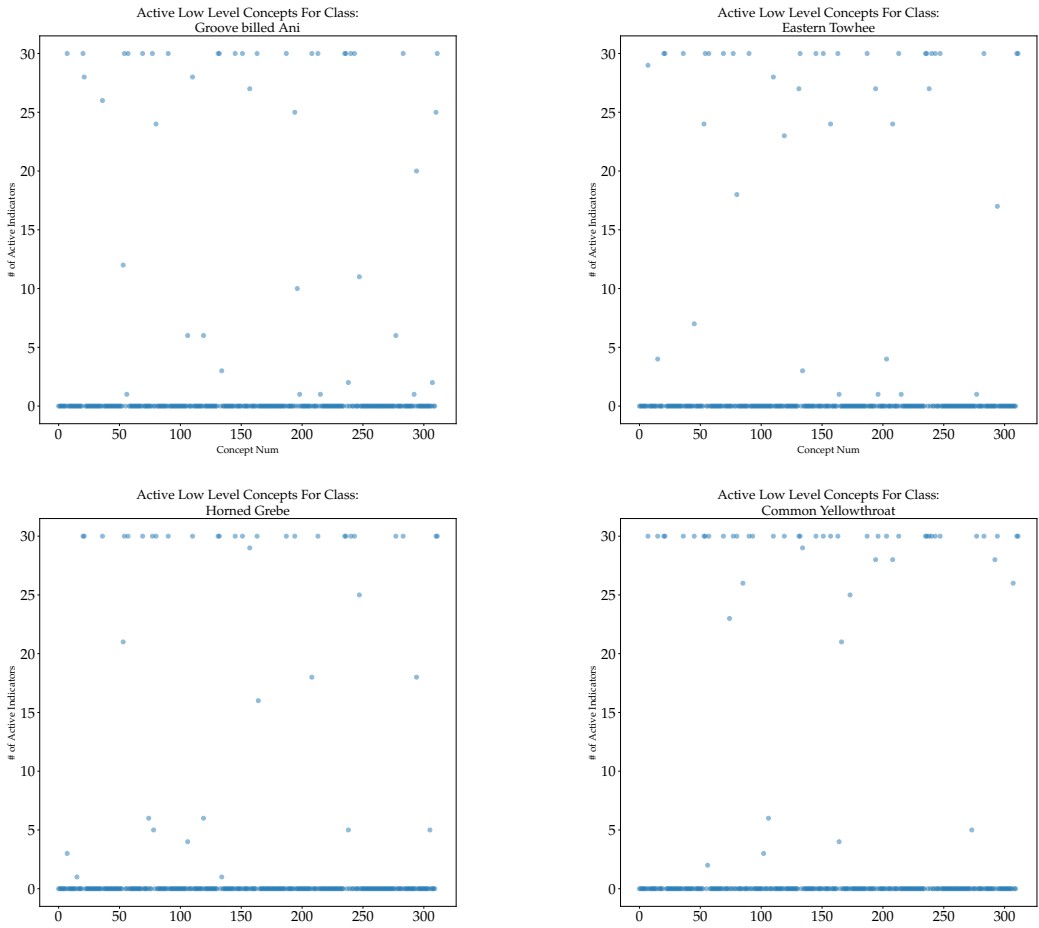

Figure 11: Summary of Activation of Low Level concepts for Classes: *Groove Billed Ani, Eastern Towhee, Horned Grebe* and *Common Yellowthroat*. In this setting, we once again observe very different patterns of concept activations for each class, albeit yielding a highly sparser representation. Similar to the High Level, within each class, there are some concepts are shared by almost all class examples as one would expect; at the same time however, we observe how the low-level concept discovery mechanism allows individuals examples to yield different activation patterns even in this highly sparse setting. This allows for the utilization of concepts that are only active in one or two examples of the class.

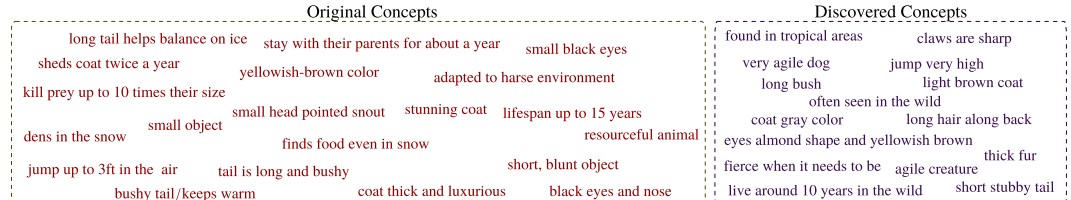

Figure 12: Original and additional discovered concepts for the *Arctic Fox* ImageNet class. By green, we denote the concepts retained from the original low-level set pertaining to the class, by maroon, concepts removed via the binary indicators $Z$, and by purple the newly discovered concepts.

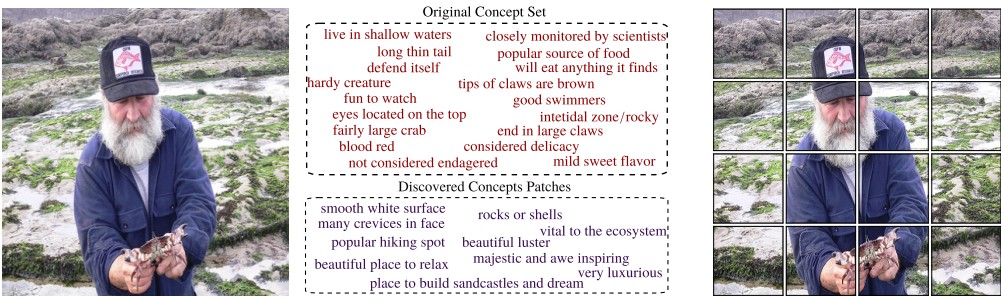

Figure 13: A random example from the *Rock Crab* class of ImageNet-1k validation set. On the upper part, the original concept set corresponding to the class is depicted, while on the lower, some of the concepts discovered via our novel patch-specific formulation.

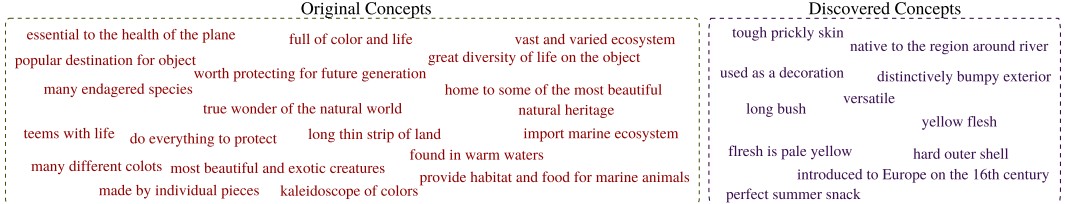

Figure 14: Original and additional discovered concepts for the *Coral Reef* ImageNet class. By green, we denote the concepts retained from the original low-level set pertaining to the class, by maroon, concepts removed via the binary indicators $Z$, and by purple the newly discovered concepts.

