# OpenReview forum: "Coarse-to-Fine Concept Bottleneck Models"
_NeurIPS.cc/2024/Conference — NeurIPS 2024 poster_

### Official Review · Reviewer_JnxP · 2024-07-06

**Soundness:** 2
**Presentation:** 3
**Contribution:** 3
**Rating:** 6
**Confidence:** 4

**Summary:**

This paper presents a type of label free concept bottleneck model for ante-hoc interpretability that incorporates a hierarchichal concept representation. The concepts are represented in a two-level hierarchy with high-level concepts denoting scenes/objects and lower level concepts denoting more specific attributes at a patch-level. Experimentally, the authors show better accuracy and better concept prediction (on datasets with GT annotations) while proposing a metric based on Jaccard index to judge quality of concept prediciton.

**Strengths:**

1. I really like the idea of hierarchichal representation of concepts. It is natural and intuitive and novel in the context of concept representations for interpretability. It's specific instantiation here is also reasonable.
2. Experiments are comprehensive in in terms of multiple, diverse large-scale datasets and baselines generally.
3. The presentation in general is strong and motivations are clear.

**Weaknesses:**

1. I have some important concerns about interpretability and soundness of label-free CBMs in general that rely on CLIP embedding similarity for concept prediction. Please see Q.1, 2 in Questions tab.

2. The only baseline I would suggest adding would be standard CBMs, specially for concept prediction accuracy (Tab. 2).
Although I expect standard CBM to perform better given its supervised training, but it'd be interesting to see how much is the performance gap if there is any.

3. It'd be interesting to see (even if qualitatively) how the system behaves with concept intervention like the original CBMs.

**Questions:**

Q.1 One key question I have is how prone is the model to detect some concept accurately but in an uninterpretable way? In other words how can we be sure the concept detection process itself is well grounded?
For eg. For a given image, is the detection of "red feather" actually because the model detects a red feather or just because of "red-colored head" of a bird? Did you try to investigate this qualitatively via some tools (maybe saliency maps or activation maximization)? How easy is it to find a poorly grounded concept detector?
Concept detection accuracy is possibly one way to evaluate this. However, a user would still need a qualitative tool for the same when the ground-truth concept annotations are not available.

Q.2 In the qualitative examples I frequently see concepts such as "become aggressive when threatened", "quick learners", which are not grounded in any visual information but our biological/behavioral/social understanding of the respective objects/classes. How do you view the use of these concepts for the purpose of image classification?

Q.3 Are the patch level CLIP embeddings extracted by resizing the patches and passing through CLIP or using patch-based embeddings of the original input image?

**Limitations:**

The authors do discuss them separately and clearly (partly in main paper and partly in appendix).

---

> ### Author Rebuttal · Authors · 2024-08-06
>
> We thank the reviewer for highlighting the strengths of our work and for raising some interesting questions.
>
> - **Weakness 2**: *Comparison to supervised baseline.*
>
> > Following the reviewer’s suggestion, and given the limited time in the rebuttal, we compare the concept prediction accuracy of our approach to [1], since they already report the performance in terms of average precision (AP) and AUC. As expected, the supervised method performs better than our approach, but not by a great margin. Specifically,  for the example-wise setting, we yield an AP of 26.90 and AUC of 67.60 The corresponding results of [1], are AP of 28.35 and AUC of 76.22 for the seen classes and AP of 25.31 and AUC of 72.10 for the unseen ones. Thus, the experimental results suggest that our framework is able to provide similar performance without the use of concept supervision.
>
> > [1] Marcos et al., Attribute Prediction as Multiple Instance Learning, TMLR, 2022
>
> - **Weakness 3**: *Test-time Interventions.*
>
> > We thank the reviewer for their suggestion. Indeed, examining the concept intervention capabilities of our framework is already part of a future work. Nevertheless, following the reviewer’s suggestion, we performed a preliminary analysis on potential interventions. In our framework, the most appropriate type of intervention corresponds to inaccurate concept activations (similar to “Type 3: Incorrect concept activations” of Label-Free CBMs). In this setting, and for each example/patch, we can readily examine and alter the concept activation patterns. To this end, we consider the example provided in Fig. 3 in the main text (and Fig. 1 in the included PDF). For this example, we intervened on the discovered concepts by randomly turning on 10 concepts that contain the words white and feathers, while randomly switching off 10 concepts that contain the word black. We performed this process ten times and observed the changes in the classification decision. Most times, i.e, 7 out of 10, we yielded classes such as quill, african grey, oystercatcher, black stork and penguin and the rest, irrelevant classes such as picket fence, can opener and geyser. We also explored a “by-hand” approach, where we switched off some concepts that were not relevant to the example in consideration, e.g., very strong structure and color depends on the type of spider, and observed no change in the classification decision. These results serve as an initial investigation in the efficiency of intervention in our framework.  We’ll add more qualitative examples in the camera-ready.
>
> - **Question 1**: *how prone is the model to detect some concept accurately but in an uninterpretable way?*
>
> > We thank the reviewer for raising this point. Indeed, one of the main motivations of this work was to avoid this kind of misdetection (L53-61) of concepts based on other information present in the image. In our work, we have access to all the concepts discovered both on the image level, but also on the patch-level. This means that we can access and visualize the discovered concepts on these two levels and examine their validity. We provide such a visualization in the included PDF. We have yet to explore other qualitative tools such as activation maximization. This is, however, a very important avenue for our future research which includes the exploration of such methods, along with the consideration of even more isolated low-level information stemming from part detection algorithms or superpixel-based approaches.
>
> - **Question 2**: *Usage of non-visual concepts.*
>
> > This is a very interesting point that the reviewer is making. This is a fair question and highlights a limitation in the existing concept datasets and particularly automatically created ones (via LLMs or other methods). Those concepts are not always strictly visual, although they may be correlated to visual concepts, making them useful for image-based classification. In the future, we plan to invest some effort in the creation of datasets with exclusively visual low-level concepts.
>
> - **Question 3**: *Patch Resizing*
>
> > The patch level CLIP embeddings are extracted by resizing first and passing through CLIP to match the standard CLIP input size. To this end, we first resize the patch to 224. We’ll include this information in the camera ready.

---

> > ### Comment · Reviewer_JnxP · 2024-08-09
> >
> > Thank you for the response. Some of my concerns have been addressed satisfactorily.
> >
> > About the concept detection visualisation (Question 1), the author response partially addresses my concern. However I still wish to enquire if in case the authors had qualitatively explored the use of saliency maps. I know a detailed study is not possible at this point but I was hoping there would be some qualitative insights.

---

> > > ### Author Response · Authors · 2024-08-09
> > >
> > > We thank the reviewer for their prompt response and we are glad we managed to address most of their concerns with our rebuttal.
> > >
> > > During the rebuttal period, we tried our best to provide some more qualitative insights in the included PDF via the visualization of patch-based concepts, which gives an indication of the spatial distribution of the detected low-level concepts.
> > >
> > > Unfortunately, we did not have time to explore any other visualization techniques such as saliency maps. This would require a thorough analysis on a per-example and per-patch basis and as the reviewer aptly notes, such a detailed study is not feasible at this point; at the same time a hasty and superficial investigation could potentially lead to imprecise results. However, the exploration of such visualization methods, along with a thorough study of the intervention behavior of our approach will be the main avenues for our future work.

---

> > > > ### Comment · Reviewer_JnxP · 2024-08-11
> > > >
> > > > Thanks for the reply. While some of my concerns remain, taking into account your rebuttal for my other concerns too, I'll raise my score to 6.

---

### Official Review · Reviewer_yu5m · 2024-07-11

**Soundness:** 2
**Presentation:** 3
**Contribution:** 3
**Rating:** 6
**Confidence:** 3

**Summary:**

The author introduced a novel Concept Bottleneck Model (CBM) that facilitates hierarchical concept learning. Specifically, the proposed Concept Discovery Block (CDB) plays a pivotal role in uncovering concepts from preprocessed image-text similarity embeddings by employing a variational Bayesian framework to learn a binary mask. Additionally, by applying the CDB module to each patch-level image to detect low-level concepts and the entire image for high-level concept discovery, information propagation between the two levels leads to robust classification performance through sparse concept learning.

**Strengths:**

- S1: As an ante-hoc interpretable CBM, the proposed method is intuitive and requires lightweight computation, which is desirable.
- S2: The performance of the proposed method is superior to other multimodal CBM baselines.

**Weaknesses:**

- W1: The class/label designated at the high level and its attributes at the low level are strong hierarchical constraints. So, the proposed method was limited to showing its applicability only in cases with a transparent hierarchical relationship between attributes and classes by specifying the pool of low-level concepts corresponding to each class. This may require burdensome human inspection to configure.
- W2: Another concern is the fixation of the interpretable threshold to all CDB modules as 0.05. The author described it as the probability value used to determine whether the specific concept is active. However, even if some image patches have the same concept, it is evident that the concept may contribute to each patch to a different extent. Therefore, dynamically adjusting or learning the threshold may perform better than a fixed threshold.

**Questions:**

Please check out the Weakness section. I listed additional questions as follows:
- Q1: For the inference phase, the author mentioned two ways to address the stochastic nature of the drawing process for the binary masks (Line 235), and the proposed method leveraged the latter one. Is there any specific reason for this? Have you compared the difference between both?

**Limitations:**

Please check out the Weakness section.

---

> ### Author Rebuttal · Authors · 2024-08-06
>
> We thank the reviewer for their time, consideration and their helpful insights that will help improve the manuscript.
>
> - **Weakness 1**: *The class/label designated at the high level and its attributes at the low level are strong hierarchical constraints.*
>
> > It is true that such a hierarchical relation between concepts is needed by our approach. We have so far performed experiments with existing benchmarks where these relations were given. In the future, we plan on expanding this to concept hierarchies that are already compiled by domain experts: for instance, organ ecosystem->plant species->traits type or disease->symptoms. Note that constructing these types of hierarchies is already common in many fields and is much less burdensome than needing to generate a dataset with per-image concept annotations. In addition, it is worth noting that, although the concept hierarchies in CUB and SUN were manually designed, in the case of the ImageNet dataset, the concepts were automatically derived via an LLM, showing that this process can be automated to some extent.
>
> - **Weakness 2**: *Another concern is the fixation of the interpretable threshold to all CDB modules as 0.05.*
>
> > We thank the reviewer for raising this interesting point that can open the way for a potential follow-up of our current approach. In this context, we would like to note that this threshold is used only during inference, while during training the model is free to adjust the “probabilities” of each concept (on either the high or the low level). Thus, during training, the model decides how useful the concept is and adjusts its probability accordingly. If during inference, the probability of activation is very small, e.g. less than 0.05 (translating to the concept being active 5 out of a 100 times), we could assume that its presence is not essential for modeling the image/patch in consideration. However, we fully agree with the reviewer that this threshold could be adjusted or learned during the process, thus resulting in a more contextualized decision in order to activate(or deactivate) concepts. For this reason, we will discuss this point as a perspective to explore in the revised version of the paper. Please also see our response to Q1 regarding an ablation study when not using a threshold.
>
> - **Question 1**: *Addressing the stochastic nature.*
>
> > The motivation behind this decision was two-fold: (i) complexity of the inference process and (ii) interpretability. Indeed, to obtain better estimates of the active concepts using sampling, we need to draw multiple samples from the learned posteriors and average the results. This increases the complexity of the approach relative to the number of inference samples. At the same time, this process renders the interpretation of the active concepts more laborious, since potentially different sets of concepts may be active per run and by averaging the results it may be more difficult for a practitioner to interpret the results. In contrast, by deterministically deciding which concepts are active via thresholding allows one to readily examine which concepts are active per example. Please also see Table 2 in the included PDF. Therein, we show that we need multiple samples to reach the performance, thus showcasing the increase in complexity and examination.

---

> > ### Comment · Reviewer_yu5m · 2024-08-12
> > **Response to the authors' rebuttal**
> >
> > Thank you for the authors' clarification and the additional experiments.
> > Since most of my concerns were addressed, I have decided to increase the score.

---

### Official Review · Reviewer_pbhW · 2024-07-12

**Soundness:** 3
**Presentation:** 3
**Contribution:** 3
**Rating:** 6
**Confidence:** 3

**Summary:**

This work introduces a novel framework that leverages recent advances in vision-language models and a Bayesian approach for coarse-to-fine concept selection. It introduces the notion of concept hierarchy, allowing high-level concepts to be characterized by lower-level attributes and exploiting granular information in image patches.

**Strengths:**

- The writing is fluent and easily comprehensible.
- Propose a novel way of assessing the interpretation capacity of CF-CBMs based on the Jaccard index between ground truth concepts and learned data-driven binary indicators.
- Extensive experiments were conducted to demonstrate that the proposed CF-CBMs outperform other state-of-the-art methods in terms of classification accuracy and interpretability.

**Weaknesses:**

- Over-reliance on the vision-language backbone's capability might result in poor performance for images from uncommon datasets.
- There is a lack of experiments on test-time concept interventions.

**Questions:**

How can it be ensured that the patches assigned to each image accurately correspond to low-level concepts?

**Limitations:**

The author mention in the Limitations of the dependence on the vision-language backbone.

---

> ### Author Rebuttal · Authors · 2024-08-06
>
> We thank the reviewer for recognizing the qualities of our approach concerning the originality, the significance, the quality and the clarity.
>
> - **Weakness 1**: *Over-reliance on the vision-language backbone's capability might result in poor performance for images from uncommon datasets.*
>
> > This is indeed a limitation that our approach shares with all methods employing VLMs, which could be alleviated by fine-tuning the VLM on a custom dataset. We will explicitly mention this in the related section.
>
> - **Weakness 2**: *There is a lack of experiments on test-time concept interventions.*
>
> > We thank the reviewer for their suggestion. Indeed, examining the concept intervention capabilities of our framework is already part of a future work. Nevertheless, following the reviewer’s suggestion, we performed a preliminary analysis on potential interventions. In our framework, the most appropriate type of intervention corresponds to inaccurate concept activations (similar to “Type 3: Incorrect concept activations” of Label-Free CBMs).
>
> >In this setting, and for each example/patch, we can readily examine and alter the concept activation patterns. To this end, we consider the example provided in Fig. 3 in the main text (and Fig. 1 in the included PDF). For this example, we intervened on the discovered concepts by randomly turning on 10 concepts that contain the words white and feathers, while randomly switching off 10 concepts that contain the word black. We performed this process ten times and observed the changes in the classification decision. Most times, i.e, 7 out of 10, we yielded classes such as quill, african grey, oystercatcher, black stork and penguin and the rest, irrelevant classes such as picket fence, can opener and geyser. We also explored a “by-hand” approach, where we switched off some concepts that were not relevant to the example in consideration, e.g., very strong structure and color depends on the type of spider, and observed no change in the classification decision. These results serve as an initial investigation in the efficiency of intervention in our framework.  We’ll add more qualitative examples in the camera-ready.
>
> - **Question 1**: *How can it be ensured that the patches assigned to each image accurately correspond to low-level concepts?*
>
> > We thank the reviewer for raising this point. Indeed, during inference, we have full access to the per-example discovered concepts encoded in the binary masks $\mathbf{Z}$; this includes both the image-level and patch-level concepts. Thus, we can visualize the results on every level and examine the behavior of the approach. Such a visualization is provided in Fig. 1 in the included PDF file.

---

> > ### Comment · Reviewer_pbhW · 2024-08-12
> >
> > I have read the author rebuttal and made any necessary changes to my review.

---

### Official Review · Reviewer_6WPf · 2024-07-12

**Soundness:** 3
**Presentation:** 2
**Contribution:** 2
**Rating:** 5
**Confidence:** 4

**Summary:**

The authors propose coarse-to-fine concept selection in Concept Bottleneck Models (CBMs). They introduce a concept hierarchy that identifies low-level concepts in local patches of input images, as well as high-level concepts in the overall images.  Additionally, the authors enhance interpretability by considering sparsity in concept predictions. Their proposed model, CF-CBM, achieves high classification performance while maintaining interpretability.

**Strengths:**

1. The authors propose a novel evaluation metric based on Jaccard similarity to evaluate concept predictions.

2. The proposed method enhances both classification accuracy and concept prediction accuracy by making predictions from local patches in a sparse manner.

**Weaknesses:**

1. While the Jaccard similarity metric effectively assesses the alignment between the model's predicted concepts and the actual concepts, it serves only as one aspect of interpretability evaluation. Notably, Table 2 indicates that the Jaccard index is quite low, raising questions about whether such a score sufficiently demonstrates the model's interpretability. Additionally, it would be helpful to clarify what factors contribute to the low Jaccard index.

2. Moreover, the concepts described in the paper appear somewhat ambiguous. I encourage the authors to refer to the Questions section for further clarification.

3. A clearer explanation is needed regarding how the authors' proposed approach enhances classification accuracy and interpretability. Specifically, it would be helpful to understand whether predicting concepts from local patches is effective, if learning class predictions aids in identifying low-level concepts, and how the application of sparsity contributes to these improvements. Additionally, an ablation study is necessary to support these claims. This also includes an explanation of why this approach can be referred to as coarse-to-fine.

**Questions:**

1. I would like to gain a clearer understanding of the high-level concepts as defined by the authors. The authors used the class name as the high-level concept set.

1-a. If the object's class is the high-level concept, can we argue that predicting the high-level concept (class) from the low-level concepts in the existing CBM does not establish a hierarchical structure between concepts?

1-b. If the object's class is indeed the high-level concept, it seems that referring to class as high-level concept may lead to confusion for readers.

1-c. When seeking low-level concepts, as the authors propose in local patches, can we always expect to find them? For instance, the characteristic "small/delicate songbird" shown in the Discovered Concepts Patches of Figure 3 does not seem to be a property that can be defined at the patch level.

2. Why are there multiple concept discovery blocks for low-level concepts?

3. Where can the discovery mentioned in lines 266-267 be found in the tables?

4. Where can the discussion in lines 341-342 be verified?

5. If low-level concepts are identified at the patch level, it seems possible to trace which patch each concept was found in. Demonstrating that the image patch where the concept was discovered aligns with the concept itself could enhance the interpretability of the proposed method.

**Limitations:**

The authors have adequately addressed limitations in the Limitations & Conclusions section. Since the proposed method uses a frozen pretrained CLIP as its backbone, the ability to discover concepts may be constrained by the limitations inherent in CLIP's training.

---

> ### Author Rebuttal · Authors · 2024-08-06
>
> We thank the reviewer for their thoughtful feedback and suggestions.
>
> - **Weakness 1**: *The use of Jaccard similarity.*
>
> > We thank the reviewer for raising this point that can help clarify the importance of using a different metric compared to the standard binary accuracy typically considered in the interpretability domain. We agree with the reviewer that the Jaccard similarity only serves as a partial investigation of a model’s interpretability evaluation. However, the emerging low score in this setting does not reduce the impact of considering this metric in the context of interpretability but instead provides a more realistic measure of the model’s capacity.
>
> > Indeed, attribute prediction is a hard multi-label problem, even for humans. Consider for instance the SUN attributes annotations; therein, every image has been independently annotated by three annotators. Out of all attributes that have been labeled as positive by at least one annotator, only 16.5% have also been labeled as positive by the other two. Evidently, considering this mismatch even between human annotators, we expect that concept-based models will exhibit difficulties in discovering concepts with or without the use of any ground truth data. This is especially true for models that additionally depend on the attribute detection capabilities of VLMs. Nevertheless,  the performance we obtain is as expected, and indeed in line with other recent work [A]. If we compare the concept prediction accuracy of our approach to [B], where the mean average precision (AP) and AUC are reported, we see that we obtain comparable results, although, as expected, the supervised method performs better than our approach by a small margin. Specifically,  for the example-wise setting, we yield an AP of 26.90 and AUC of 67.60 The corresponding results of [B], are AP of 28.35 and AUC of 76.22 for the seen classes and AP of 25.31 and AUC of 72.10 for the unseen ones. Thus, the experimental results suggest that our framework is able to provide similar performance without the use of concept supervision. Yet, we agree that this has potential implications for the interpretability of any CBM approach, although studying this falls beyond the scope of this work.
>
> >[A] Rao, S., et al. "Discover-then-Name: Task-Agnostic Concept Bottlenecks via Automated Concept Discovery." arXiv preprint arXiv:2407.14499 (2024).
>
> >[B] Marcos et al., Attribute Prediction as Multiple Instance Learning, TMLR, 2022
>
> - **Weakness 2**: *Ablations and clarifications*
>
> > We thank the reviewer for their comments that will help clarify the performed ablation studies, while improving the work with additional results. We’ll clearly state the settings in the experimental section to avoid any potential confusion.
>
> > **Effectiveness of predicting concepts from local patches.**
> Predicting low-level concepts from the patches themselves (without first considering a high level concept detection) is denoted in Table 1 as $CDM^L$, with and without sparsity, while in Table 2, $CDM^L$ denotes using the patches and the low-level concepts with the sparsity mechanism (and again without high level concept detection). In all cases, we observe that predicting concepts solely using patches and low-level concepts, and without any other mechanism, yields subpar performance both classification-wise but also in terms of attribute matching. In contrast, when we add the high-level concept detection process and tie the levels together, the classification and attribute matching performance significantly improve.
>
> > **Learning class predictions (High-level Concepts) aids in identifying the low-level concepts.**
> This setting indeed corresponds to our CF-CBM method, where we discover the high-level concepts, and then discover the low level ones. From Table 2, we observe that our method significantly improves the concept prediction capabilities of CBMs, improving both on the $CDM^L$ method discussed previously and the CDM method that in this setting considers the whole image and the low-level concepts. Thus the experimental results suggest that learning the high level predictions and propagating this information to the patch level, significantly improves the attribute prediction capabilities of the emerging model.
>
> > **How the application of sparsity contributes to this improvement.**
> We thank the reviewer for this suggestion, since it will help enrich the results of the work. In Table 1 in the main text, we present $CDM^H$ and $CDM^L$ with and without the sparsity mechanism. Therein, we observe that when using the sparsity inducing mechanism, we obtain on par or even improved classification performance while using only a subset of the concepts.
> As far as the concept detection capabilities are concerned, if we remove the sparsity mechanism, we are essentially using all the concepts, and we thus lose the ability to explicitly assess which concepts are active to assess the results.
>
> > However, following the reviewer’s proposal, we introduce a novel ablation study, where we vary the Bernoulli prior probability $\alpha$, that directly affects the obtained sparsity. The higher the value of alpha, the less sparse are the obtained results; this allows for assessing the impact of the sparsity inducing behavior of the proposed model as per the reviewer’s suggestion. The obtained results are depicted in Table 1 in the included PDF. In summary, we observe that the sparser the representation, the better the attribute matching capabilities of the model. These results also highlight the necessity of another metric apart from the classification accuracy to assess the interpretation capabilities of the resulting models, since all settings exhibit similar classification performance. We’ll add these results in the camera-ready.

---

> ### Author Response · Authors · 2024-08-06
>
> - **Question 1**: *Clearer understanding of the high-level concepts.*
>
> > We thank the reviewer for this point, that will help clarify any potential confusion concerning our construction.  Our approach requires a pre-existing concept hierarchy to model high and low level concepts. The underlying assumption is that attributes/low-level concepts will generally describe a partial view of the object, often relevant to a certain part of the object or its environment, while the class/high-level concepts tends to describe the entirety of the object, thus requiring access to the full image. This makes the patch-based representation more appropriate for the low-level concepts and the image representation for the high-level concepts.
>
> > To this end, we indeed use class labels as high-level concepts, and attribute labels (CUB, SUN) or more descriptive concepts (ImageNet) as low-level concepts. In this context,  the class labels of ImageNet and SUN highly reflect the main object/scene in each image, having sufficiently descriptive class names such as abbey, airport, black swan, etc. On the other hand, the low level concepts represent more specific descriptions of the objects in consideration, such as red bill, black feathers, bricks, etc.
> Our approach can be extended to deeper hierarchies (e.g. scene-object-part-subpart) and this is part of our future work.
>
> - **Question 1a**: *CBM and hierarchical structure.*
>
> > Indeed, this can be said from other CBMs. The main difference between our approach and other CBM models is that, in ours, the inference of the high-level concept is performed twice: first with a regular model that has access to the whole image, whose output conditions the patch-based model that predicts the low-level concepts. This allows the low-level concept predictor to be guided by whole-image information in an interpretable manner via the high-level concept predictor. These low-level concepts are then themselves used to predict, once again, the high-level concept in an interpretable way. In contrast to this, other CBMs perform only the latter step.
>
> > With respect to the class being used as the high-level concepts, it  is important to clarify that in our work, an example can have multiple high-level concepts (that do not necessarily have to be class names). These are not set or fixed; the prediction of these is based on both: (i) the VLM similarity between the high-level concepts and the image, and  (ii) the discovery mechanism that aims to learn which concepts are essential for modeling the given example. Thus, we are not just setting the high level concept to be the class name. The high-level concepts are discovered via the described process; these are then used towards classification.The same holds for the low-level concepts.
>
> - **Question 1b**: *If the object's class is indeed the high-level concept, it seems that referring to class as high-level concept may lead to confusion for readers.*
>
> > Please see our response to the previous question. We do not explicitly set a single class name as the high level concept. An example can have multiple high-level concepts that are discovered by the described mechanism and that do not necessarily have to be the class names. Our approach can consider any kind of high-level concepts; however, this kind of “class-attributes” structure is the most common setting in existing datasets and thus was chosen for our experiments. We’ll clarify this in the camera-ready to avoid any potential confusion.
>
> - **Question 1c**: *When seeking low-level concepts, as the authors propose in local patches, can we always expect to find them?*
>
> > Indeed, depending on how each specific dataset has been designed, not all low-level attributes will be appropriate for a patch-level representation. Our approach alleviates this by making the low-level concept prediction aware of, not just the image patches, but also an interpretable whole-image representation in the form of the high-level concept predictions. We would like to note that some of the concepts, particularly in ImageNet and SUN, cannot be considered to be visual in nature, meaning that the VLM will only be able to capture them via visual correlations with the non-visual concept.
>
> - **Question 2**: *Why are there multiple concept discovery blocks for low-level concepts?*
>
> > Since each patch is treated as a standalone image, the concept discovery block needs to be applied per patch. We will rework Figure 1 to make this clearer.
>
> - **Question 3**: *Where can the discovery mentioned in lines 266-267 be found in the tables?*
>
> > We apologize for the typo in this case; the discovery corresponds to the sparsity entry in the Table. We’ll correct the error in the camera-ready.

---

> ### Author Response · Authors · 2024-08-06
>
> - **Question 4**: *Where can the discussion in lines 341-342 be verified?*
>
> > After training, and for each example,we have access to the activated high and low level concepts. Thus, using these, we can construct a per class summary of activations as described in the main text. To this end, we considered the concept activation patterns for all the examples in the Sussex Spaniel class, and observed the described behavior.
>
> - **Question 5**: *If low-level concepts are identified at the patch level, it seems possible to trace which patch each concept was found in.*
>
> > We thank the reviewer for raising this point. Indeed, during inference, we have full access to the per-example discovered concepts encoded in the binary masks Z; this includes both the image-level and patch-level concepts. Thus, we can visualize the results on every level and examine the behavior of the approach. Such a visualization is provided in Fig. 1 in the included PDF file.

---

> ### Comment · Reviewer_6WPf · 2024-08-11
>
> Thank you for your response. The author’s response helped clarify some confusing aspects and addressed some of my concerns. However, it is a weakness that the discussion on concept hierarchy remains focused on class and low level concepts. While the authors mentioned the introduction of the notion of concept hierarchy as a novel contribution, the hierarchy between class and low-level concepts has already been explored in existing CBM frameworks. Although the authors suggests that high-level concepts can be defined beyond just classes through rebuttal, this claim has not been validated within the scope of this study. As a result, only the identification of low-level concepts at the patch level is recognized as a contribution.

---

> > ### Author Response · Authors · 2024-08-11
> >
> > We thank the reviewer for the appreciation of our work and our rebuttal.
> >
> > Our intention with this work is to introduce a method that is suitable for deeper concept hierarchies through the usage of dependent concept sets (e.g. object categories->attributes) and multi-level representations (e.g. whole images->patches). To this end, we considered pairs of [whole images, high-level concepts] and [patches, low-level concepts]. In this context, we explored how each model behaves, independently or connected to the other level, while introducing an evaluation metric of interpretability. During the rebuttal, we also had the opportunity to enrich our manuscript with suggestions of new ablation studies and qualitative evaluations.
> >
> > Nevertheless, due to constraints in the available concept datasets, we acknowledge that our experimental setup is currently limited to the usage of class names as high level concepts. For this reason, we agree that it would be suitable to reformulate the main novelty of our approach as "using the detection of high-level concepts, such as object or scene categories, to provide context for the detection of low-level concepts, such as object attributes".
> > We will focus the writing on this aspect, for which we have provided substantial experimental evidence, and relegate the more general concept hierarchy formulation to the future work section.
> >
> > Finally, we believe our efforts address most of the concerns raised, and we kindly ask the reviewer to take them into consideration in the final score associated to our submission.

---

> > > ### Comment · Reviewer_6WPf · 2024-08-12
> > >
> > > After considering your response and other reviews, I have decided to increase the score.

---

### Author Rebuttal · Authors · 2024-08-06

We thank all the reviewers for taking the time to review our manuscript and for their insightful comments.

We carefully considered all the comments that the reviewers raised and addressed them diligently. To this end, we respond to each question individually and we also include a PDF with some novel investigations, both qualitative and quantitative, based on the feedback and comments of the reviewers.

---

> ### Comment · Area_Chair_Aajt · 2024-08-08
> **Reviewer-author discussions**
>
> To the authors:
> Thanks a lot for the rebuttal.
>
> To all reviewers:
> Please read the rebuttal and other review comments if necessary, and have a discussion with the authors if you have further comments.
>
> Best,
> AC

---

> ### Author Response · Authors · 2024-08-13
>
> We would like to thank again the AC and the reviewers  for their thoughtful engagement and the discussions throughout this discussion period.  We are committed to refining our paper in alignment with their suggestions.

---

### Decision · Program_Chairs · 2024-09-25

**Decision:**

Accept (poster)

**Comment:**

Initially, this paper received mixed ratings: 4, 7, 5 and 5. After post-rebuttal discussion, the ratings were changed to 5, 6, 6, and 6: three upgraded and one downgraded but still positive. Three reviewers upgrading the ratings were satisified with rebuttal. The reviewer downgrading the rating did not provide detailed comments to the rebuttal, but still kept positive. The AC read the comments and the paper carefully.